# Lifting Traces to Logic: Programmatic Skill Induction with Neuro-Symbolic Learning for Long-Horizon Agentic Tasks

Jie-Jing Shao [1 2]   Haiyan Yin [2]   Yueming Lyu [2]   Xingrui Yu [2]   Lan-Zhe Guo [1 3]
Ivor W. Tsang [2]   James T. Kwok [4]   Yu-Feng Li [1 5]

## Abstract

Foundation model-driven agents often struggle with long-horizon planning due to the transient nature of purely prompting-based reasoning. While existing skill induction methods mitigate this by distilling experience into state-blind parameterized scripts, they fail to capture the conditional logic required for robust execution in dynamic environments. In this paper, we propose Neuro-Symbolic Skill Induction (NSI), a framework that lifts interaction traces into modular, *logic-grounded* programs. By synthesizing explicit control flows and dynamic variable binding, NSI empowers agents to discover *when* and *why* to act. This paradigm enables the efficient generalization, allowing agents to induce skills from few-shot examples and flexibly adapt to unseen goals. Experiments on a series of agentic tasks demonstrate that NSI consistently outperforms state-of-the-art baselines, empowering agents to self-evolve into architects of logic-grounded skills. Project Page: https://sh-jj.github.io/NSI.

## 1. Introduction

The rapid progress of large language models (LLMs) has empowered agentic systems to perceive, reason, and act in complex environments (Yao et al., 2022b; Shinn et al., 2023; Jimenez et al., 2024). However, general-purpose foundation models often lack grounded knowledge for specific real-

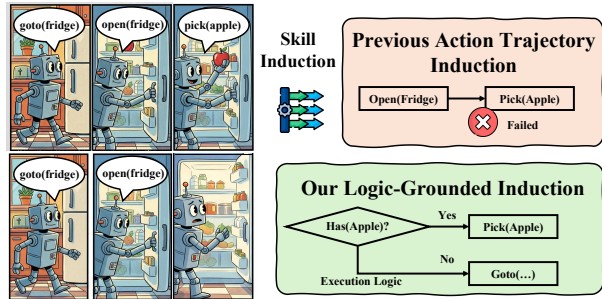

Figure 1. **From Trace Scripts to Logic-Grounded Programs.** Existing methods induce skills as *state-blind* parameterized scripts, often failing when environmental deviations occur. In contrast, **NSI** lifts traces into logic-grounded workflows. By explicitly synthesizing state predicates and control flow (e.g., branching logic), our framework empowers agents to improve generalization.

world domains. Consequently, they exhibit domain-specific reasoning gaps, leading to unreliable tool use and failures in long-horizon planning (Zhang et al., 2025c; Jiang et al., 2025; Zhou et al., 2026; You et al., 2026; Jia et al., 2026a). These limitations highlight that static pre-trained models are insufficient. Instead, an agentic system must self-evolve through environmental interaction to bridge the gap between general capabilities and task requirements.

A representative paradigm for such evolution is skill induction (Wang et al., 2025c;b). Unlike episodic-memory-based evolution (Shinn et al., 2023; Wang et al., 2024b) that relies on textual retrieval to a laborious process of re-reasoning, skill induction leverages the interactive nature of agentic tasks to extract reusable, executable procedures. The learned skills expand the action space with high-level skills that can be directly invoked without re-planning (Wang et al., 2025b; Zheng et al., 2025a; Yu et al., 2025). This process mirrors the cognitive transition from System-2 to System-1: by consolidating deliberate reasoning chains into fast, reflexive "muscle memory" (skills), agents effectively internalize experience as an executable ability (Anderson, 1982; Singley & Anderson, 1989). However, current methods typically induce skills as *parameterized scripts (e.g., Open(Receptacle) → Pick(Object))*. These induced skills produce state-blindly actions, and thus struggle to faithfully

[1]State Key Laboratory of Novel Software Technology, Nanjing University, China [2]Centre for Frontier AI Research, and Institute of High Performance Computing, Agency for Science, Technology and Research, Singapore [3]School of Intelligence Science and Technology, Nanjing University, China [4]Department of Computer Science and Engineering, Hong Kong University of Science and Technology, China [5]School of Artificial Intelligence, Nanjing University, China. Correspondence to: Haiyan Yin <yin_haiyan@cfar.a-star.edu.sg>, Yu-Feng Li <liyf@nju.edu.cn>.

*Proceedings of the $43^{rd}$ International Conference on Machine Learning*, Seoul, South Korea. PMLR 306, 2026. Copyright 2026 by the author(s).

represent the underlying execution logic. As shown in Fig. 1, after *Open(Fridge)*, the ideal execution logic is to check if the desirable object, 'apple', is present. If so, dynamically bind it to *Pick(Object)*; if not, branch to search elsewhere.

In this paper, we propose NSI, a neuro-symbolic skill induction framework that bridges this gap by redefining skills as modular, logic-grounded programs. NSI employs a trace-to-logic lifting mechanism that abstracts raw interaction histories into control workflows, where actions are produced by explicit symbolic predicates. By modularizing execution logic and enabling dynamic variable binding, NSI empowers agents to transcend rigid script memorization. This paradigm shift enables the self-evolution to robustly adapt to the environments with varying topologies and recursive complexities. Our contributions are summarized as follows:

**Logic-Grounded Modular Representation.** We propose a neuro-symbolic representation that decouples neural perception from symbolic execution. Unlike state-blind scripts, our skills invent explicit control flows and dynamic variable bindings grounded in First-Order Logic, enabling rigorous state-awareness and modular composition.

**Trace-to-Logic Induction Framework.** We introduce a framework that lifts instances into generalized programs. Guided by an *empirical consistency* objective, it progressively consolidates local experts into global skills via iterative refinement, maximizing the empirical coverage.

**Continuous Self-Evolution via Reflective Planning.** We introduce a mechanism converting runtime failures into skill honing. By grafting successful recovery trajectories onto failure nodes, agents continuously "grow" their skill graphs. This turns transient errors into permanent capabilities, yielding state-of-the-art performance across diverse benchmarks.

## 2. Related Work

### 2.1. Skill Discovery in Traditional RL

Learning specialized skills for sequential decision-making tasks has been developing for a long time in traditional reinforcement learning (RL). They introduce the skills or options (Sutton et al., 1999) to enable temporal abstraction by encapsulating primitive actions into reusable, temporally extended behaviors, and enhancing policy generalization (Pertsch et al., 2021; Nam et al., 2022; Shi et al., 2023). However, these methods face challenges when applied to foundation model-driven agents. First, the learned skills are black-box neural policies, which are incompatible with the interpretable, text-based reasoning of LLMs. Second, they rely on extensive parameter optimization, which is impractical for large-scale foundation models due to sample inefficiency and high computational cost. NSI retains the motivation of temporal abstraction, but shifts skill acqui-

sition from policy optimization to LLM-guided program induction, allowing reusable skills to be distilled from a few demonstrations without retraining opaque policies.

### 2.2. Programmatic Skill Induction in the LLM Era

Programmatic Skill Induction has emerged as a promising paradigm for the evolution of foundation model-driven agents. Unlike neural parameters which are opaque and computationally expensive to update, it abstracts reusable procedures from experience (Wang et al., 2025c) and organizes sequential actions into executable skills (Wang et al., 2025b; Zheng et al., 2025a; Yu et al., 2025). Adjacent studies further explore reusable task knowledge through inheriting generalizable knowledge from LLMs, integrating task-specific experts, and learning procedural memory from experience (Liu et al., 2026; Lin et al., 2026; Mi et al., 2026). In this paradigm, programs serve as a natural representation that is both synthesizable and understandable by foundation models, transforming programs into a functional medium for skill induction and execution. This alignment enables agents to accumulate experience and evolve capabilities through abstraction and induction, with recent work further studying open-world fragility and structural failure refinement of agentic skills (Wu et al., 2026; Li et al., 2026). Nevertheless, the sample efficiency of program induction does not by itself solve robustness: most induced programs still behave as parameterized action traces, reusing *what* to do while leaving state conditions and argument-resolution logic implicit. The remaining gap is representational: skills must encode the execution logic that makes a procedure valid, rather than merely compress successful traces.

### 2.3. Automated Agentic Workflow Generation

This work is aligned with the field of Agentic Workflow Generation (Zhang et al., 2025b; Qiao et al., 2025; Wang et al., 2025a), which aims to automate the construction and optimization of structured reasoning graphs. We adopt the paradigm of modular workflow representation, leveraging evolutionary mechanisms to induce and refine skill structures (Hu et al., 2025; Zhang et al., 2025a; Shang et al., 2025; Niu et al., 2025; Zheng et al., 2025b; Gungordu et al., 2026). Unlike existing approaches that organize predefined reasoning nodes (e.g., Debate, Ensemble) and focus on prompt or topological optimization, we introduce a fundamental shift towards logic-grounded skill induction. Our method invents control nodes and variable-binding nodes to organize primitive action nodes, fundamentally determining *"When"* and *"Why"* based on state grounding. By compiling experiences into executable skills, NSI moves workflow structure from a meta-level reasoning scaffold into the skill itself, where control and binding decisions are grounded in environment state. This differs from structured reasoning methods that improve LLM reasoning through fixed high-level modeling

templates, multi-agent debate over symbolic translations, or world-model-based planning (Yang et al., 2026; Xiong et al., 2026; 2025): workflow generation shows the value of structure, while NSI targets the internal execution logic that makes a reusable skill robust under state variation.

### 2.4. Neuro-Symbolic Learning for Agentic Tasks

Neuro-symbolic learning explores how to combine traditional symbolic reasoning with data-driven learning to enhance models' generalization and reliability (Mao et al., 2025; Yang et al., 2025). In sequential decision-making, neuro-symbolic representations ground learned behavior in structured state-action spaces: relational states and action models represent object-centric planning problems (Xu et al., 2019; Garrett et al., 2021), symbolic abstractions connect observations with parameterized skills in grounded imitation (Shao et al., 2025a), and verification-oriented representations avoid reasoning shortcuts or infeasible programs (Yang et al., 2024; Jia et al., 2025). As agentic tasks increasingly rely on foundation models, the same representational need reappears at higher-level components of agency: goal representations make compositional language instructions explicitly verifiable (Shao et al., 2026); reasoning representations support self-evaluation, open-world logical reasoning, proactive control, or domain-specific abstraction (Shao et al., 2025b; Mo et al., 2026; Xiang et al., 2026; Wu et al., 2025; Yu et al., 2026b; Jia et al., 2026b); and action-interface representations improve open-world tool use and web control by structuring actions, feedback, and feasibility contracts (Wu et al., 2026; Yu et al., 2026a; Cai et al., 2026; Xiang et al., 2026).

Recent agentic systems induce reusable skills from experience (Wang et al., 2025c;b). NSI extends this skill-induction setting by making the *skill representation itself* neuro-symbolic: neural grounding binds predicates and arguments to observations, while a symbolic execution graph specifies applicability conditions, dynamic argument binding, control-flow branches, and recovery logic. This makes induced skills state-conditioned and inspectable rather than state-blind action scripts.

## 3. Preliminaries

**Problem Formulation.** Following (Wang et al., 2025b; Zheng et al., 2025a; Yu et al., 2025), we define the goal-conditioned POMDP planning problem fromulation $\mathcal{M} = (\mathcal{S}, \mathcal{S}^+, \mathcal{A}, \mathcal{T}, \mathcal{G})$, with raw state space $\mathcal{S}^+$, observation space $\mathcal{S}$, actions ($a \in \mathcal{A}$), transition function $\mathcal{T} : \mathcal{S} \times \mathcal{A} \to \mathcal{S}$ and goal $g \in \mathcal{G}$. Following (Xu et al., 2019; Garrett et al., 2021; Mao et al., 2025), we focus on object-centric modeling: states are described as relational features among entities in the environment, i.e., a set of task-related objects $\mathcal{O}$, actions are defined functions that take entity names as inputs

and can be executed in the environment. $\mathcal{P}$ is a set of task-related predicate symbols , a ground atom $p$ is a predicate that contains specific arguments, such as Contains(bridge, apple). If a state $s$ satisfies $s \models p$, it indicates that $s$ semantically entails the interpretation of $p$. The agentic task is to generate an execution trajectory $\tau = (s_0, a_0, \ldots, s_H)$ that terminates in a state satisfying the overall task goal $g$.

**LLM-Driven Skill Induction.** To bridge the gap between interaction experiences and reusable capabilities, the community introduce the concept of *programmatic skills* (Wang et al., 2024a; 2025b). Generally, it is defined as an executable program that composes a sequence of primitive actions to achieve a parameterized sub-goal $\omega(\theta)$, where the parameterizers $\theta$ represent the different instantiations of the sub-goal $\omega$, e.g., *find_and_pick(desirable_item='apple')*. Unlike black-box neural policies, these skills are represented as code artifacts (e.g., Python functions), allowing LLMs to read, write, and execute them directly. Formally, given a dataset $\mathcal{D}_\omega = \{\tau_1, \ldots, \tau_N\}$ of successful trajectories $\tau_i$ satisfying $\omega(\theta_i)$ (i.e., $s_{H_i} \models \omega(\theta_i)$), the objective is to induce a skill $\pi_\omega$ that maximizes execution success:

$$\pi_\omega^* = \underset{\pi_\omega \sim P_{\mathrm{LLM}}(\cdot | \mathcal{D}_{\pi_\omega})}{\arg\max} \sum_{\tau_i \in \mathcal{D}_\omega} \mathbb{I}\left[\mathrm{Exec}(\pi_\omega, \mathrm{Init}(\tau_i)) \models \omega(\theta_i)\right]$$

where $P_{\mathrm{LLM}}$ denotes the probability sampling by LLMs, and $\mathbb{I}[\cdot]$ is the indicator function. The term $\mathrm{Exec}(\pi_\omega, \mathrm{Init}(\tau_i)) \models \omega(\theta_i)$ signifies that executing the program $\pi_\omega$ starting from the initial state of $\tau_i$ results in a state satisfying the sub-goal $\omega(\theta_i)$. Existing works (Wang et al., 2025c) employ an LLM-as-judge to estimate the success rate on $\tau_i$, while other approaches assume the states from empirical trajectories could be re-instantiated and verify the proposed skill $\pi_\omega$ in the online environment (Wang et al., 2025b; Zheng et al., 2025a; Yu et al., 2025; Shi et al., 2026). Once validated, the induced skills are registered into a *Skill Library*, expanding the action space with reliable sub-routines. By prioritizing these stable skills over iterative reasoning, agents mitigate cascading errors in long-horizon tasks.

## 4. A Neuro-Symbolic Skill Representation

In this section, we propose a *Neuro-Symbolic Representation* that bridges the gap between observations and actions with neural perception and symbolic execution, transforming skills from static scripts into dynamic, logic-grounded workflows. Departing from rigid parameterized scripts, our formalism leverages explicit First-Order Logic (FOL) and graph-structured control flows to enable verifiable correctness, dynamic variable binding, and modular composition.

### 4.1. Skill Workflows with Neuro-Symbolic Grounding

A fundamental challenge in agentic skill learning lies in bridging the gap between unstructured, ambiguous obser-

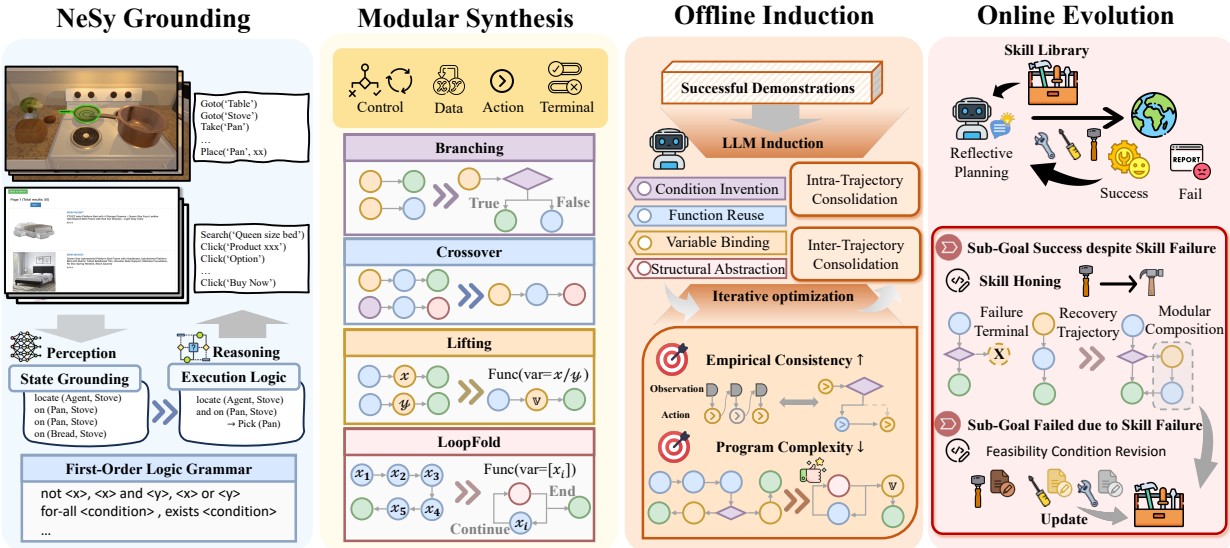

*Figure 2.* The overall framework of NSI. Starting with NeSy Grounding, the system maps environmental perception to a logical exuction space governed by First-Order Logic. The core mechanism, Offline Induction, abstracts successful demonstrations into reusable skills via modular synthesis. These skills are maintained in a library and updated through Online Evolution, where a reflective planner utilizes interaction feedback to correct feasibility conditions and modularly compose recovery trajectories to hone the skills.

vations and the precise, state-dependent logic required for robust skill execution. To address this, we propose a **Neuro-Symbolic Skill Representation** that explicitly decouples *perceptual grounding* from *execution workflow*. Concretely, we formalize a skill schema as a triple $\pi_\omega = (\theta_\omega, \phi_\omega, G_\omega)$, consisting of invocation parameters $\theta_\omega$, a neural grounding module $\phi_\omega$, and a symbolic execution graph $G_\omega$. This architecture assigns distinct operational roles to leverage the complementary strengths of neural and symbolic systems:

**Neural Feedback Perception ($\phi_\omega$):** We employ the LLM as a *semantic grounding perceptor*. At each step, it processes raw, unstructured environmental observations $s_t$ to update a structured *symbolic state* $Z_t$. This acts as the bridge: since the program's execution logic relies on symbolic predicates, this transformation provides the necessary *symbolic basis* for the program to interpret diverse environmental feedback and precisely determine the subsequent action.

**Symbolic Execution Logic ($G_\omega$):** The core behavioral logic is encapsulated into a workflow graph $G_\omega$, which is executed by a *symbolic interpreter*. Once synthesized, this graph dictates the control flow via *conditional determinism*. By operating on the grounded symbolic predicates in $Z_t$, it employs explicit branching to rigorously map diverse state configurations to specific actions. This effectively extracts a reusable and adaptive execution workflow. Unlike prior state-blind scripts that follow rigid sequences, our logic-grounded graph naturally incorporates environmental feedback into the control structure, enabling the skill to

autonomously adapt to diverse state configurations.

### 4.2. Inducing Execution Logic via Node Invention

The symbolic workflow $G_\omega = (V, E)$ is structured as a directed graph sharing an internal scope $\mathcal{C}$. This scope acts as a centralized memory for invocation parameters and intermediate variables, which nodes $v \in V$ consume and update via defined interfaces. Unlike conventional workflow generation methods (Zhang et al., 2025b;a; Shang et al., 2025; Zheng et al., 2025b) assembling pre-defined nodes (e.g., Debate, Voting), we constructs skills by **inventing the internal execution logic** of each node. Nodes are not static pre-defined tools, but generative programmatic primitives synthesized to govern specific aspects of execution. In this section, we focus on the functional semantics of these modules, deferring the induction details to Sec. 5. Specifically, we categorize them into four logic-grounded operators:

**Data Operation (*DataOp*): Inventing Dynamic Variable Binding.** *DataOps* represent the *reasoning* steps that precede action, which synthesizes an explicit variable transformation program $f_v : \mathcal{C} \times \mathcal{Z} \to \mathcal{C}$ that queries the *symbolic world state* $\mathcal{Z}$ using variables from the current *scope* $\mathcal{C}$ and writes the resolved result back to $\mathcal{C}$. This enforces a *think-then-act* logic, ensuring necessary arguments are rigorously resolved before execution. To achieve this, given a vocabulary of task-related predicates (e.g., *locates, contains*) and logical grammar (e.g., *and*, *or*, *select_one*), the framework adaptively composes these primitives to invent specific

binding expressions (e.g., *target=select_one(x, is_type(x, 'apple') ∧ contains(loc, x))* where *loc=select_one(y, locates('agent', y)))*), rather than invoking pre-defined rules. This expressive composition allows the agent to articulate complex state dependencies and autonomously discover the precise variables needed for robust execution.

**Control Operation (*CheckOp, LoopOp*): Inventing Decision Boundaries.** These nodes govern the flow of execution, determining *when* to branch or iterate, which synthesizes an explicit conditional expression $p_v : \mathcal{C} \times \mathcal{Z} \rightarrow \{True, False\}$ that captures the execution branch. By utilizing the same predicate vocabulary and logical grammar, the framework composes discriminative logical expressions (e.g., constructing *is_closed(y) ∧ locates('agent',y)* as the critical condition) to capture the execution branch. This explicitly integrates environment information into the control flow, enabling the skill to adapt to dynamic state changes.

**Primitive Operation (*PrimitiveOp*): Grounded Action Execution.** *PrimitiveOps* serve as the interface to the environment, which executes atomic actions $A(var)$ parameterized by the shared scope $\mathcal{C}$. Instead of using static values, the arguments *var* are dynamic references explicitly bound by preceding *DataOps* (e.g., *pick(target)* where *target* is dynamically resolved from $\mathcal{C}$). This ensures that physical actions are strictly grounded in the preceding reasoning steps, guaranteeing that the agent's behavior is rigorously aligned with its logical derivation.

**Terminal Operation (*TerminalOp*): Execution Feedback.** These nodes mark the conclusion of a skill's lifecycle. A *Success* node returns a completion signal, validating the goal. A *Failure* node notifies the agent of termination while automatically generating a logic-grounded diagnosis based on the execution context (e.g., returning *"Target not found after navigation: ∄ x, is_type(x, 'apple')"*). This feedback attributes the failure of the skill execution logic in potential new instances, facilitating effective re-planning.

This fine-grained, modular node design is pivotal for localized refinement. By encapsulating logic into distinct, invented units, the agent can iteratively refine specific nodes (e.g., correcting a binding logic in a *DataOp*) without needing to regenerate the entire skill, enabling efficient induction.

### 4.3. Interactive Execution Semantics with Control Flow

Execution is formalized as a dynamic traversal over $G_\omega$, maintained by an explicit state $\xi_t = (v_t, \mathcal{C}_t, Z_t)$, structured to strictly separate control logic from perceptual grounding:

**Neural State Tracking (Perceive):** Instead of conflating perception and execution, during the skill execution, the LLM functions as a semantic state tracker $\phi_\omega$, parsing $o_{t+1}$ to update specific predicates in $Z_t$ (e.g., updating 'is_open(x)' when the observation feedbacks).

**Symbolic Resolution (Think):** The interpreter deterministically evaluates node $v_t$ based on $\mathcal{C}_t$ and $Z_t$, performing variable binding or branching (e.g., evaluating 'is_open(x)') to determine the next step, ensuring traceable control flow. It emits a null action $a_t = \perp$, preserving $o_{t+1} = o_t$.

**Grounded Interaction (Act):** At a *PrimitiveOp*, the system emits a physical action $a_t \neq \perp$, whose arguments are precisely instantiated from $\mathcal{C}_t$, triggering an environment transition and obtaining a new observation $o_{t+1} = \mathcal{T}(o_t, a_t)$.

## 5. Logic-Grounded Skill Induction

Having established the neuro-symbolic representation of skills as executable graphs $G_\omega$ in Section 4, we now address the algorithmic challenge of synthesizing these structures from interaction experiences.

To bootstrap the induction, we first extract sub-goals from the raw interaction trajectories. Similar to (Wang et al., 2025c;b; Zheng et al., 2025a; Yu et al., 2025), we utilize the LLMs to identify resuable process and critical state changes in the raw trajectories, effectively segmenting long-horizon episodes into sub-goal directed sub-trajectories $\tau^\omega$. We assume access to a dataset $\mathcal{D}_\omega = \{(\tau_i^\omega, \omega(\theta_i))\}$ of such segmented demonstrations for the each sub-goal $\omega(\theta)$.

### 5.1. Empirical Programmatic Consistency

In partially observable environments such as embodied or web tasks, perfectly re-instantiating the environment to verify a skill hypothesis is often infeasible. Therefore, instead of online environment verification (Wang et al., 2025b; Yu et al., 2025; Zheng et al., 2025a), we ground our consistency check in the historical execution traces. We verify whether the induced skill logic, when executed against the recorded states, faithfully reproduces the expert's action sequence while accommodating necessary internal reasoning steps. We define *empirical programmatic consistency* by synchronizing the skill's action steps with the expert's trajectory. A skill $\pi_\omega$ instantiated with $\theta$ at step $h$ is consistent with trace $\tau \in \mathcal{D}_\omega$ starting at $h$ if its generated actions $\hat{a}$ match the expert's $a^*$ starting from $s_h$:

$$\text{Consistent}(\tau, \pi_\omega, \theta, h)$$
$$= \mathbb{I}\left[\forall k, \hat{a}_k \neq \perp \implies \hat{a}_k = a^*_{h+m(k)}\right] \quad (1)$$

where $m(k)$ maps the $k$-th program step to the corresponding action index. Formally, we try to synthesize a skill policy $\pi_\omega$ that maximizes the empirical consistency of the data $\mathcal{D}_\omega$ while minimizing program complexity. We define the *Empirical Consistency Region* $\widehat{\mathcal{R}}_{\pi_\omega}^\tau$ as the set of states in a trajectory $\tau$ from which $\pi_\omega$ can successfully reproduce

the subsequent expert actions. The induction objective is:

$$\max_{\pi_\omega} \sum_{\tau \in \mathcal{D}_\omega} \left| \widehat{\mathcal{R}}^\tau_{\pi_\omega} \right| - \lambda \left| \pi_\omega \right| \tag{2}$$

$$\widehat{\mathcal{R}}^\tau_{\pi_\omega} \triangleq \left\{ s_h \in \tau \mid \exists \theta, \texttt{Consistent}(\tau, \pi_\omega, \theta, h) = 1 \right\}.$$

where $|\pi_\omega|$ denotes the program complexity. This objective encourages the discovery of skills that are both empirically consistent and structurally concise.

### 5.2. Progressive Skill Induction from Experiences

Given the combinatorial and non-differentiable nature of the program space $\mathbb{G}$, directly maximizing the global objective in Eq. 2 is intractable. To address this, we propose a progressive specialization-to-generalization process: (1) Intra-Trajectory Consolidation, which synthesizes Local Experts $\pi_\tau$ specifically fitting each demonstration trace, and (2) Inter-Trajectory Merging, which recursively unifies these local experts and constructs a generalized skill program $\pi_{\text{global}}$ that satisfies the Minimum Description Length principle. In this subsection, we simplify the notation $\pi_\omega$ as $\pi$, as the induction process operates on a given sub-goal $\omega$.

**Stage 1: Intra-Trajectory Logic Consolidation.**

The first stage aims to synthesize a *local expert* $\pi_\tau$ that maximizes consistency within a single reference trajectory $\tau \in \mathcal{D}_\omega$. This ensures that the skill logic is at least locally valid, and serves as an initialization for the optimization. We employ an iterative *consistency check and refinement* loop. The system scans the trajectory for an *uncovered state* $s_{\text{err}}$ (where $s_{\text{err}}$ is a state within the sequence $\tau$) for which the current policy $\pi_\tau$ fails the consistency check (i.e., $s_{\text{err}} \notin \widehat{\mathcal{R}}^\tau_{\pi_\tau}$). This is analogous to identifying hard negatives or counter-examples in program synthesis. Upon detection, the synthesizer updates $\pi_\tau$ to resolve the conflict at $s_{\text{err}}$ by computing a *Structural Consolidation* of the logic. This typically involves introducing conditional branches (*CheckOp*) to handle the divergence, effectively expanding the skill's feasibility region. Critically, this allows the system to discover latent branching logic (e.g., "open door only if closed") even from a single linear trace by identifying state conditions that necessitate different actions, thereby improving generalization.

**Stage 2: Inter-Trajectory Skill Consolidation.**

The second stage maximizes the global empirical consistency (Eq. 2) by consolidating multiple local experts into a generalizable global skill. We adopt a greedy iterative optimization. At each iteration $k$, we first identify the *hardest constraint* $\pi_{\text{hard}}$, the local expert corresponding to the trajectory $\tau$ that is least covered by the current global skill $\pi^k_{\text{glb}}$. We then synthesize a candidate skill $\pi^{k+1}_{\text{cand}}$ by structurally

merging the current global skill with this bottleneck expert:

$$\pi^{k+1}_{\text{cand}} = \texttt{Consolidate}(\pi^k_{\text{glb}}, \pi_{\text{hard}}) \tag{3}$$

The global skill is updated to $\pi^{k+1}_{\text{cand}}$ only if the candidate satisfies *Feasibility Dominance* (i.e., strictly expanding coverage while maintaining consistency):

$$\pi^{k+1}_{\text{glb}} \leftarrow \begin{cases} \pi^{k+1}_{\text{cand}}, & \text{if } \widehat{\mathcal{R}}_{\pi^{(k)}_{\text{glb}}} \subset \widehat{\mathcal{R}}_{\pi^{\text{cand}}} \\ \pi^{(k)}_{\text{glb}}, & \text{otherwise} \end{cases} \tag{4}$$

We implement the `Consolidate` operator via LLM-based program synthesis, which introduces abstract variables and control flows to unify distinct logical paths, utilizing the MDL principle to prevent overfitting.

**Structural Operators for Logic Consolidation.**

To effectively navigate the program space towards the MDL-optimal solution in both stages, we equip the LLM synthesizer with a set of primitive structural operators. These operators are designed to resolve logical conflicts via branching or compress redundancy via abstraction:

**Conditional Branching** (`Branching`): Aligning two program graphs to find their common structure. For divergent paths, it resolves conflicts via *Predicate Invention*: synthesizing a discriminative predicate (e.g., inserting `if is_closed(fridge)` to unify a trace that opens the fridge with one that does not) to act as a guard condition. This explicitly introduces new control flow to handle environmental variations, enabling *state-dependent execution* that dynamically adapts to the current context.

**Modular Crossover** (`Crossover`): Exploits the modularity of the graph representation to inherit structural components. Because each node (e.g., a *CheckOp* or a specific action sub-routine) acts as an independent module defined by its variable interface, this operator can identify useful subgraphs in a donor skill $\pi_{\text{hard}}$ and graft them into $\pi_{\text{glb}}$. The adaptation is achieved solely by rebinding the input parameters of the transferred nodes to the execution context of $\pi_{\text{glb}}$, enabling efficient logic reuse without re-synthesis.

**Variable Lifting** (`Lifting`): Converts instance-specific traces into generalized logic by lifting hard-coded constants to explicit input parameters. Instead of binding to specific entities (e.g., one trace washes an apple at a `sink` and another uses a `basin`), the operator prompts the LLM to discover the shared *permutation-invariant property* of the entities (e.g., being a washable station) across traces and exposes it as a parameter. This empowers the agent to autonomously select any valid resource at runtime (e.g., choosing sink, basin or other washable station), achieving generalization across functionally equivalent objects.

**Loop Folding** (`LoopFold`): Synthesizes iterative control flow by detecting repetitive substructures acting on different

*Table 1.* **Main Results.** We compare NSI against baselines on ALFWorld, WebShop, and TextCraft. We report Success Rate for all domains, and additionally Reward Score for WebShop. Our proposed method achieves consistent improvements.

| Method | ALFWorld Success Rate (%) | WebShop Score(%) | WebShop Success Rate (%) | TextCraft Success Rate (%) |
|---|---|---|---|---|
| ReAct (Yao et al., 2022b) | 85.8 | 44.0 | 20.0 | 62.0 |
| Reflexion (Shinn et al., 2023) | 84.3 | 40.8 | 23.0 | 59.0 |
| ADaPT (Prasad et al., 2024) | 67.9 | 45.8 | 29.0 | 72.5 |
| StateAct (Rozanov & Rei, 2025) | 84.3 | 40.6 | 8.00 | 83.0 |
| AWM$_{offline}$ (Wang et al., 2025c) | 88.6±1.1 | 51.8±1.7 | 32.2±1.3 | 66.3±2.0 |
| AWM (Wang et al., 2025c) | 91.3±0.8 | 49.2±1.9 | 30.0±2.0 | 92.5±3.6 |
| ASI (Wang et al., 2025b) | 70.6±1.9 | 7.7±1.7 | 7.50±3.0 | 77.8±1.8 |
| NSI w.o. online honing | 93.5±1.9 | 58.8±1.8 | 30.5±1.5 | 78.5±2.5 |
| NSI (Ours) | **98.0±1.2** | **76.5±1.2** | **44.5±1.5** | **95.2±0.8** |

targets. The operator abstracts a sequential unrolling (e.g., check(A), check(B), ...) into a concise *LoopOp* that iterates over a given list (derived from an external argument or an internal *DataOp*). This enables the skill to scale to variable-sized object collections while maintaining a compact program representation regardless of the number of targets, adhering to the MDL principle.

### 5.3. Online Skill Evolution via Reflective Planning

Following standard skill-based planning frameworks (Wang et al., 2024a; 2025c), we utilize the LLM as a planner that dynamically selects between invoking an induced skill or executing a raw action. The skills are exposed to the planner via their function signatures and docstrings, which summarize the *Feasibility Region* $\widehat{\mathcal{R}}_\sigma$, allowing the LLM to gauge applicability based on the current state description.

**Reflective Planning with State-aware Feedback.** The proposed logic-grounded skills offer more than binary success signals. When a skill execution fails, it terminates at a specific *Failure Node* that returns diagnostic symbolic feedback (e.g., the fridge is still closed and cannot sure objects inside, *is_closed(fridge)*). It provides the reasons of execution failure, enabling *Reflective Planning*, utilizing this diagnostic information to generate a corrective plan, comprising alternative skills, or raw actions (e.g., *open(fridge)*), to bridge the gap from the failure state to the original sub-goal.

**Skill Honing.** This reflective mechanism closes the loop for self-evolution, driving the continuous refinement of $\pi_\omega$ based on the *outcome of the recovery attempt*. If the corrective plan succeeds, the system treats the recovery trajectory as a newly discovered logical branch. It employs *Logic Consolidation* operators to "grow" the skill graph by grafting this logic onto the failure node, effectively generalizing the skill to handle previously unmodeled exceptions via new conditional paths. This new subgraph is initially marked as *tentative* to maintain policy stability, and is permanently solidified only if it consistently resolves the failure in sub-

sequent encounters. Conversely, if the recovery fails, it signifies that the sub-goal is fundamentally unattainable from the current state. The system then restricts the skill's usage by updating its docstring to explicitly exclude such infeasible contexts, thereby aligning the skill's description with its actual feasible region.

## 6. Empirical Study

### 6.1. Experimental Setup and Baselines

The evaluation is conducted on three agentic benchmarks:

**ALFWorld** (Shridhar et al., 2021) is a text-based embodied environment derived from the ALFRED (Shridhar et al., 2020) 3D household robotics simulator. The benchmark evaluates agents on long-horizon decision-making under partial observability, comprising 134 test instances across six task types. Following established protocols (Yao et al., 2022b), we utilize the two standard demonstrations per task type, previously used for in-context prompting, as the data source for offline skill induction.

**WebShop** (Yao et al., 2022a) simulates a rich e-commerce platform. The benchmark assesses an agent's ability to navigate product pages, choose attributes, and execute purchases to satisfy user queries. To ensure fair comparison, we restrict skill induction to a single successful purchase trajectory, aligning with the one-shot demonstration setup used in standard in-context learning baselines.

**TextCraft** (Prasad et al., 2024) evaluates compositional generalization in Minecraft crafting on 200 recursive tasks (depths 2–4). With only simple demonstrations (3 expert trajectories, depth=1), agents must compose primitives to solve unseen long-horizon goals.

**Competing Baselines.** We compare our approach against a diverse set of baselines representing different agentic paradigms: (1) **ReAct** (Yao et al., 2022b), a well-validated prompting strategy that interleaves reasoning traces with

action execution. (2) **Reflexion** (Shinn et al., 2023), which leverages linguistic episodic memory to reflect on past failures and improve future performance. (3) **ADaPT** (Prasad et al., 2024), which improves long-horizon planning by explicitly decomposing overall tasks into sub-goals. (4) **StateAct** (Rozanov & Rei, 2025), which guides the agent to perform explicit state analysis before determining actions. (5) **AWM** (Wang et al., 2025c), which abstracts reusable workflow patterns from successful trajectories into text-based memory. (6) **ASI** (Wang et al., 2025b), which induces parameterized programmatic skills but lacks the explicit logic-grounded control flow and variable binding of our method. For fair comparison, we utilize GPT-4o as the backbone model across all methods and set the temperature to 0 to improve the reproducibility. We report mean $\pm$ standard deviation over 3 runs for skill-induction methods.

## 6.2. Main Results

We present the results in Table 1. Our proposed NSI consistently achieves consistent performance improvements across three benchmarks, surpassing the strongest baselines by significant margins. From the results, we could find that:

**Limitations of Unstructured Memory and Pure Decomposition.** The performance of Reflexion indicates that linguistic episodic memory offers limited generalization across varying episodes, yielding negligible gains over ReAct. While ADaPT excels in recursive decomposition for TextCraft, it shows limited effectiveness in ALFWorld. In such domains, the difficulty arises from partial observability and long-horizon interaction required to realize sub-goals, rather than high-level decomposition. Consequently, ADaPT's focus on planning fails to address the critical bottleneck of long-horizon execution.

**The Necessity of Expressive Skill Representation.** AWM achieves stable gains by abstracting experiences into text-based workflows, validating the importance of *process reuse*. However, when ASI attempts to formalize these into parameterized programs, performance drops. This reveals an *expressiveness gap*: linear scripts cannot capture the complex logic inherent in the tasks, rendering them less effective than even unstructured text summaries.

**Advantage of Logic-Grounded Skills.** In contrast, our NSI leverages neuro-symbolic learning to generate *logic-grounded* skills. Unlike linear scripts, NSI's programs internalize reusable processes into verifiable, state-aware logic (e.g., branches and loops). This capability significantly enhances the agent's ability to realize sub-goals, driving consistent and substantial improvements in both long-horizon execution and generalization.

To understand why NSI outperforms baseline programmatic agents, we analyze the structural properties of the induced

skills and their impact on long-horizon planning.

**Efficiency and Logical Cohesion.** We first investigate the internal execution efficiency of individual skills. As shown in Figure 4a, NSI exhibits a significant advantage in handling extended interaction sequences. Specifically, while maintaining a high sub-goal success rate, NSI achieves a 100%–140% improvement in the average number of atomic steps (*avg_steps*) executed per skill invocation compared to ASI. This indicates that our logic-grounded representation possesses higher semantic abstraction and stronger logical cohesion. By internalizing complex state-dependent logic (e.g., conditional recovery and dynamic binding) within the skill itself, NSI prevents the agent from losing sight of long-term goals during execution, a common failure mode for linear agents in dynamic environments.

**Quantifying the Horizon Gap.** A survival analysis (Figure 4b) reveals a critical divergence at the 22-step threshold. While baselines match NSI on shorter tasks, they suffer a *Long-Horizon Collapse* ($> 22$ steps) due to accumulating reasoning errors, dropping to zero success. In contrast, NSI sustains performance about 50 steps. By encapsulating $\sim$7.4 atomic actions per skill, NSI effectively compresses the planning horizon, shifting from fragile micro-management to robust macro-execution.

**Horizon Compression via NSI.** Conversely, NSI successfully bridges this gap, sustaining valid execution up to 53+ steps without a corresponding spike in high-level planning complexity. By encapsulating approximately 7.4 atomic steps into each modular skill, NSI effectively "compresses" the planning horizon back into the LLM's comfort zone. This shift from micro-management to macro-execution enables NSI to solve tasks that are mathematically intractable for linear script-based agents, transforming transient runtime interactions into stable, goal-directed abilities.

## 6.3. Modular Skill Honing

Figure 3 illustrates the dynamic *modular graph growth* of skills during online deployment. Initially, the skill functions as a logic skeleton induced from minimal offline data. However, as the agent interacts, runtime failures inevitably expose logic gaps, i.e., states where the initial logic is insufficient. Instead of discarding the faulty skill, *Reflective Planning* catches the failure and generates a local corrective trajectory. This trajectory is subsequently synthesized into a functional subgraph and *grafted* onto the specific *Failure Node* via topological expansion. This process transforms terminal failures into conditional recovery branches, enabling the skill to incrementally "grow" to handle emerging edge cases (e.g., adding a `CheckOp` for closed doors). Consequently, the skill evolves from a simple linear chain into a robust, branching network, repairing its own logic without altering the correctly functioning parts.

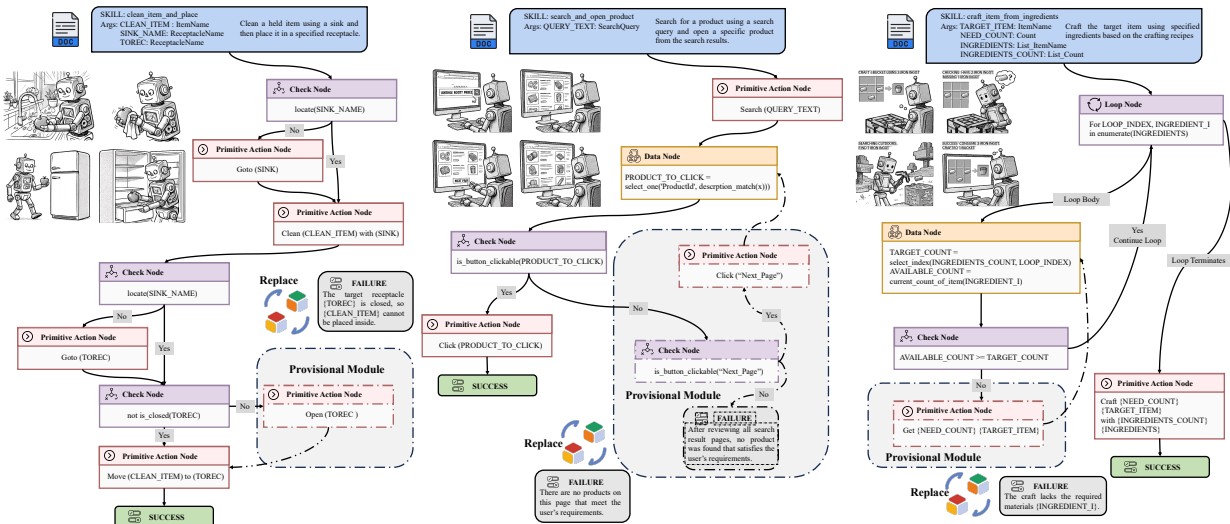

Figure 3. Representative cases of online skill evolution across three benchmarks.

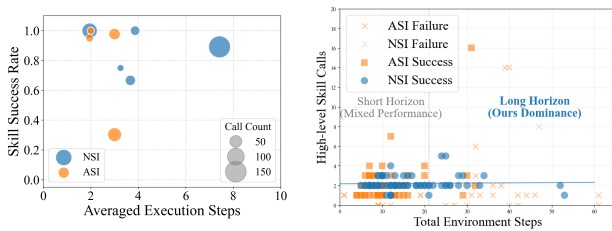

*(a)* Execution Efficiency     *(b)* Long-Horizon Robustness

Figure 4. **Impact of Logic-Grounded Skills.** (a) NSI encapsulates complex logic into multi-step skills, efficiently compressing the planning horizon. (b) This structural advantage prevents the execution collapse observed in baselines during long-horizon tasks.

## 7. Conclusion

In this paper, we introduced Neuro-Symbolic Skill Induction (NSI), a framework that transforms transient agentic reasoning into persistent, logic-grounded skills. By lifting interaction traces into modular programs, NSI transcends the limitations of linear parameterized scripts, synthesizing explicit control flows and dynamic variable bindings. Furthermore, our proposed Reflective Planning mechanism empowers agents to autonomously repair and grow these skill graphs during deployment, converting runtime failures into permanent logic improvements. Empirical results across embodied, web, and game domains demonstrate that NSI significantly compresses planning horizons and achieves efficient generalization across tasks and domains.

One interesting direction is automated predicate discovery, easing the hand-crafted predicate bottleneck while probing oracle, learned, noisy, or missing interfaces. Another is continual task learning, where modular functions and skills can be reused and evolved across related tasks.

## Acknowledgements

This research was supported by the Jiangsu Science Foundation (BK20232003,BK20243012,BG2024036), Natural Science Foundation of China (62576162), the Fundamental Research Funds for the Central Universities (022114380023) and the National Research Foundation, Singapore under its AI Singapore Programme (AISG Award No: AISG-NMLP-2024-003).

Yueming Lyu is partially supported by Career Development Fund (CDF) of the Agency for Science, Technology and Research (A*STAR) (No: H26-KSR0056), and the National Research Foundation, Singapore and Infocomm Media Development Authority under its Trust Tech Funding Initiative. Any opinions, findings and conclusions or recommendations expressed in this material are those of the author(s) and do not reflect the views of the National Research Foundation, Singapore and Infocomm Media Development Authority.

## Impact Statement

This work introduces NSI, a framework that enhances long-horizon planning and generalization for LLM-driven agents. By lifting traces into logic-grounded programs, NSI promotes resource-efficient AI development and reduces dependence on extensive demonstrations or manual engineering. While these advances improve agents in complex digital and embodied environments, more capable autonomous agents may also be misused in sensitive domains. In high-impact settings, learned skills should remain inspectable and subject to human oversight, so reuse does not obscure harmful assumptions or failures. Such methods should therefore be paired with safety alignment and ethical guardrails.

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

*Table 2.* The domain-independent first-order logic language, a general reasoning language across domains.

| Name | Signature | Meaning |
|------|-----------|---------|
| variables | $x, y, z, \cdots$ | Variables that refer to items in the planning domain. |
| types | $Object, Num, Bool, List\_(type)$ | The types of objects, numbers, boolean values, and lists of those types. |
| *Condition Operators* | | |
| predicate | $p(x, y) \rightarrow Bool$ | The function that takes objects and numbers and returns boolean values. |
| not | $not\ BoolExpr \rightarrow Bool$ | The negation of a boolean expression. |
| and, or | $BoolExpr_1 and\ BoolExpr_2 \rightarrow Bool$ | The conjunction/disjunction of a boolean expression. |
| $<, >, ==,$ | $NumExpr_1 < NumExpr_2 \rightarrow Bool$ | The comparison of two numerial expressions. |
| *Variable-Binding Operators* | | |
| attribute | $count(x) \rightarrow Num$ | The function that takes a object and returns its attribute, e.g. the count. |
| +-*/ | $NumExpr_1 + NumExpr_2 \rightarrow Num$ | The numerical calculation of two numerial expressions. |
| select_idx | $s_{idx}(List, NumExpr) \rightarrow Object$ | Return the variable at the given index from a ordered list. |
| select_one | $s_{one}(List, BoolExpr) \rightarrow Object$ | Return a object satisfying the given boolean expression. |
| select_all | $s_{all}(List, BoolExpr) \rightarrow List$ | Return all objects satisfying the given boolean expression. |

# A. The Representation Language of NSI

In this section, we provide a detailed formal specification of the Domain Specific Language (DSL) used to represent the logic-grounded skills in NSI. The representation relies on a First-Order Logic (FOL) grounded vocabulary to bridge the gap between neural perception and symbolic execution.

## A.1. Syntax

The core syntax of our logic-grounded language is summarized in Table 2. The language consists of two primary layers:

1. **Expression Layer:** Used within nodes to query states and derive variables. This primarily includes First-Order Logic predicates (e.g., is_open$(x)$) and set operations (e.g., select_one).

2. **Control Layer:** Specifies the workflow structure, including conditional branching and loops, which are encoded directly into the graph topology of $\pi_\omega$.

## A.2. Semantics of Skill Nodes

The execution of a skill $\pi_\omega$ is governed by the traversal of its graph $G_\omega$, where each node type defines specific semantic operations on the shared program scope $\mathcal{C}$ and symbolic state $\mathcal{Z}$.

### 1. Data Operation (*DataOp*):
*Semantics:* A DataOp node $v$ acts as a variable binding unit. It executes a synthesized function $f_v : \mathcal{C} \times \mathcal{Z} \rightarrow \mathcal{C}$ that resolves abstract arguments into concrete entities.
*Execution Logic:*

- **Input:** Takes current scope $\mathcal{C}$ containing unbound or abstract variables.

- **Process:** Evaluates a logic expression (e.g., $target = $ select_one$(Os, $ is_apple$(x) \wedge $ is_clean$(x))$). This imposes a "Think" step, ensuring actions are parameterized by state-verified entities.

- **Output:** Updates $\mathcal{C}$ with the resolved variable 'target'.

### 2. Control Operation (*ControlOp*):
*Semantics:* A ControlOp node functions as a decision gate, evaluating a boolean predicate $p_v : \mathcal{C} \times \mathcal{Z} \rightarrow \{\top, \bot\}$ to determine the flow of control.
*Execution Logic:*

- **Condition Evaluation:** The node checks a state condition (e.g., is_closed$(target)$).

*Table 3.* The task-specific predicates defined for ALFWorld, WebShop, and TextCraft. These predicates are used to ground the observation into symbolic states.

| Name | Signature | Meaning |
|---|---|---|
| *ALFWorld (Embodied Household)* | | |
| is_type | $is\_type(x, T) \rightarrow Bool$ | Check if object $x$ belongs to category $T$. |
| is_clean | $is\_clean(x) \rightarrow Bool$ | Check if object $x$ is in a clean state. |
| is_hot | $is\_hot(x) \rightarrow Bool$ | Check if object $x$ is heated. |
| is_cool | $is\_cool(x) \rightarrow Bool$ | Check if object $x$ is cooled. |
| is_open | $is\_open(x) \rightarrow Bool$ | Check if receptacle $x$ is open. |
| contains | $contains(x, y) \rightarrow Bool$ | Check if receptacle $x$ contains object $y$. |
| holds | $holds(x) \rightarrow Bool$ | Check if the agent is currently holding object $x$. |
| locates | $locates(x) \rightarrow Bool$ | Check if agent is at $x$. |
| *WebShop (E-Commerce)* | | |
| if_button_clickable | $if\_clickable(b) \rightarrow Bool$ | Check if UI element $b$ is visible and clickable. |
| descrption_match | $match(x) \rightarrow Bool$ | Check if search result $x$ matches non-price constraints. |
| price_match | $price\_match(x) \rightarrow Bool$ | Check if search result $x$'s price satisfies the budget. |
| current_price_match | $curr\_price() \rightarrow Bool$ | Check if the currently open item satisfies the budget. |
| current_descrption_match | $curr\_match() \rightarrow Bool$ | Check if the open item matches goal constraints. |
| current_option_match | $opt\_match(o) \rightarrow Bool$ | Check if required option $o$ has been correctly selected. |
| current_option_clicked | $opt\_clicked(o) \rightarrow Bool$ | Check if option $o$ has been clicked on the current page. |
| *TextCraft (Minecraft Crafting)* | | |
| goal | $goal(x, n) \rightarrow Bool$ | Check if the goal requires $n$ units of item $x$. |
| inventory | $inventory(x, n) \rightarrow Bool$ | Check if the agent has $n$ units of item $x$. |
| unavailable | $unavailable(x) \rightarrow Bool$ | Check if item $x$ is currently unavailable. |
| current_count_of_item | $count(x) \rightarrow Count$ | Return the current quantity of item $x$ in the inventory. |

- **Branching:**

  - If $\top$ (True): Transition to the "Then" branch (e.g., `open`($target$)).
  - If $\bot$ (False): Transition to the "Else" branch (e.g., skip to `pick`($target$)).

- This mechanism enables the skill to explicitly handle environmental variations (e.g., open/closed doors) without hallucinating actions.

**3. Primitive Operation (*PrimitiveOp*):**
*Semantics:* Represents the atomic interface to the environment.
*Execution Logic:* Invokes the underlying environment API with arguments strictly fetched from $\mathcal{C}$ (e.g., `Env.step(pick,` $id = target.id$`)`).

**4. Terminal Operation (*TerminalOp*):**
*Semantics:* Marks the end of execution and returns a status signal $S \in \{$*Success*, *Failure*$\}$. In case of failure, it returns a symbolic diagnosis (e.g., `ERR: Argument 'target' not found`), which drives the Reflective Planning process.

## B. Experimental Details

All baselines utilize **GPT-4o** as the backbone LLM with temperature set to 0.0 to improve the reproducibility.

To ensure reproducibility, we provide the key system directives used in NSI. Full prompts will be available after the code release.

---

**OBSERVATION TO PREDICATES PROMPT**

You are an AI embodied agent. Your task is to convert each observation (`observation: ...`) into grounded facts for the predicates defined below, based on the literal semantics of each predicate. For each observation:
1. Analyze the text and determine which predicate instances are explicitly supported or falsified. 2. Output two blocks:

---

- `add_facts`: List every predicate instance that is definitively true after reading the observation.

- `remove_facts`: List predicate instances that the observation falsifies (e.g., an item was previously on a table but is now described elsewhere). If none, return `[]`.

Use the canonical predicate names and argument order:

- `locates(receptacle_id)`: True when the observation states (explicitly or implicitly) that the agent is at/on/inside the receptacle. Accept cues such as "You are at ..." or "On the X you see ..." (which implies the agent is at X). If the agent moves, include the previous location in `remove_facts`.

- `reachable(receptacle_id)`: True when the observation says the agent can reach/go to the receptacle or when the environment lists nearby surfaces. Treat co-listed surfaces as mutually reachable unless specified otherwise.

- `is_open(receptacle_id)`: True only if the observation states that the receptacle is open. False if it states the receptacle is closed/shut.

- `is_closed(receptacle_id)`: True if the observation states that the receptacle is closed/shut, false otherwise.

- `contains(receptacle_id, item_id)`: True when the observation enumerates items on/in the receptacle. If an item is removed, include the previous `contains` relation in `remove_facts`.

- `holding(item_id)`: True if the observation states "You pick up ...," "You are holding ...," or similar. False if the observation says you dropped or placed it elsewhere.

- `is_cleaned(item_id)`: True when the observation states that the item is cleaned, false otherwise.

- `is_cooled(item_id)`: True when the observation states that the item is cooled, false otherwise.

- `is_heated(item_id)`: True when the observation states that the item is heated, false otherwise.

- `is_turned_on(item_id)`: True when the observation states that an item (e.g., a lamp) is turned on, false otherwise.

Formatting rules:

- Convert names to `snake_case` with numeric suffixes (e.g., "sofa 1" → `sofa_1`).

- Use exact predicate syntax, e.g., `locates(sofa_1)`.

- List only facts newly entailed or falsified by this observation; do not repeat persistent facts.

- If no changes are entailed, return `"add_facts": []` and/or `"remove_facts": []`.

Example:
Input: facts: [] action: observation: You are in the middle of a room. Looking quickly around you, you see a coffeetable 1, a diningtable 1, a drawer 4, a drawer 3, a drawer 2, a drawer 1, a dresser 1, a garbagecan 1, a sidetable 2, a sidetable 1, and a sofa 1. Your task is to: put two cellphones on the sofa.
Output:

```
{
  "add_facts": [
    "locates(middle_of_room)",
    "reachable(coffeetable_1)",
    "reachable(diningtable_1)",
    "reachable(drawer_4)",
    "reachable(drawer_3)",
    "reachable(drawer_2)",
    "reachable(drawer_1)",
```

```
      "reachable(dresser_1)",
      "reachable(garbagecan_1)",
      "reachable(sidetable_2)",
      "reachable(sidetable_1)",
      "reachable(sofa_1)"
    ],
    "remove_facts": []
}
```

Input: facts: [ ... ] action: goto dresser 1 observation: On the dresser 1, you see a book 2 and a mug 1.
Output:

```
{
  "add_facts": [
    "locates(dresser_1)",
    "contains(dresser_1, book_2)",
    "contains(dresser_1, mug_1)"
  ],
  "remove_facts": ["locates(middle_of_room)"]
}
```

Next input: facts: [ ... ] action: take mug 1 from dresser 1 observation: You pick up the mug 1 from the dresser 1.
Output:

```
{
  "add_facts": [
    "holding(mug_1)"
  ],
  "remove_facts": [
    "contains(dresser_1, mug_1)"
  ]
}
```

Follow this schema strictly for each observation.
facts: {facts}
action: {new_action}
observation: {new_observation}

---

## SUBGOAL EXTRACTION PROMPT

You are an AI planner for an embodied environment. Summarize the sub-goals from the provided interaction trajectories. These sub-goals should capture the overall task goal and the intermediate steps to achieve it.
# Data
Here are some interaction trajectories:
{examples}
# Detailed Instructions for Sub-Goal Generation:

- **Complexity**: Sub-goals should have mediocre complexity:
  - Must include at least two primitive actions; sub-goals implementable by a single action OR one condition check followed by one primitive action are NOT allowed.
  - Steps list length must be between 2 and 5 inclusive (reject 1-step designs).

- **Generality**: Sub-goals must be applicable to other similar tasks, not just the provided examples.

- **Arguments**: Use common variable types (e.g., strings, lists). Avoid complex inputs like another function.

- **Parameters and Types**: For each sub-goal, explicitly list the input parameters and their types.

- – Allowed parameter types (choose only from): `ReceptacleName`, `ItemName`, `ReceptacleType`, `ItemType`, `List_T` (e.g. `List_ItemName`, `List_ReceptacleType`).
- – Typed placeholder examples (primitive actions for illustration):
  - * go to {{`gotoRec: ReceptacleName`}}
  - * open {{`openRec: ReceptacleName`}}
  - * take {{`takeItem: ItemName`}} from {{`fromRec: ReceptacleName`}}
  - * cool {{`coolItem: ItemName`}} with {{`recep: ReceptacleName`}}
  - * heat {{`heatItem: ItemName`}} with {{`recep: ReceptacleName`}}
  - * clean {{`cleanItem: ItemName`}} with {{`recep: ReceptacleName`}}
  - * move {{`moveItem: ItemName`}} to {{`toRec: ReceptacleName`}}
  - * use {{`desklamp: ItemName`}}
- – Typed placeholder examples (overall goal for illustration):
  - * 'find an item of type {{`itemType: ItemType`}} and put it in a receptacle of type {{`RecType: ReceptacleType`}}'
  - * 'put an item of type {{`itemType: ItemType`}} on a receptacle of type {{`RecType: ReceptacleType`}}'
  - * 'clean an item of type {{`itemType: ItemType`}} and put it in a receptacle of type {{`RecType: ReceptacleType`}}'
  - * 'put a clean item of type {{`itemType: ItemType`}} in a receptacle of type {{`RecType: ReceptacleType`}}'
  - * 'put a hot item of type {{`itemType: ItemType`}} in a receptacle of type {{`RecType: ReceptacleType`}}'
  - * 'heat an item of type {{`itemType: ItemType`}} and put it in a receptacle of type {{`RecType: ReceptacleType`}}'
  - * 'put a cool item of type {{`itemType: ItemType`}} in a receptacle of type {{`RecType: ReceptacleType`}}'
  - * 'cool an item of type {{`itemType: ItemType`}} and put it in a receptacle of type {{`RecType: ReceptacleType`}}'
  - * 'examine an item of type {{`itemType: ItemType`}} with the desklamp'
  - * 'look at an item of type {{`itemType: ItemType`}} under the desklamp'
  - * 'put two items of type {{`itemType: ItemType`}} in a receptacle of type {{`RecType: ReceptacleType`}}'
- – In the JSON output, provide parameters as strings in the form `"param_name: param_type"`.

- **Start Conditions (Preconditions)**: For each sub-goal, specify clear, checkable preconditions under which the sub-goal can begin. If these are not satisfied, this sub-goal MUST NOT be selected.

  - – Express conditions succinctly with parameter references (e.g., "the agent does not locate at {{`gotoRec`}}, and {{`gotoRec`}} is reachable", "the agent already locates at {{`openRec`}} and {{`openRec`}} is current closed", "the agent already holds a {{`moveItem`}} and locates at {{`toRec`}} and the {{`toRec`}} is current open").
  - – Prefer condition forms that can be grounded in observations or known state (e.g., `reachable`, `is_open`/`is_closed`, `contains`, `holding`, `exists`).
  - – Keep preconditions minimal and necessary; avoid hidden assumptions.

- **Success Conditions (Postconditions)**: For each sub-goal, specify the observable state that must hold after successful completion. If these are not satisfied, the sub-goal is incomplete/failed.

  - – Use the same expression style as `start_conditions` with parameter references (e.g., "the agent locates at {{`gotoRec`}}", "{{`openRec`}} is current open", "{{`toRec`}} contains {{`moveItem`}}", "the agent holds a {{`moveItem`}}").
  - – Keep postconditions minimal, necessary, and directly verifiable from observations or world facts.

- **Naming**: Name each sub-goal to summarize a multi-step intent, not a single primitive action.

  - Be concise and descriptive; use lowercase with underscores.
  - Do NOT reuse primitive action names or their near variants: `go_to`, `open`, `take`, `put`, `move`, `clean`, `heat`, `cool`, `use`.
  - Prefer names like: `locate_and_open_receptacle`, `retrieve_item_from_container`, `transfer_item_to_surface`, `clean_item_and_place`, `heat_item_then_place`.

- **Steps Analysis**: For each sub-goal, describe the sequence of actions required to achieve it.

  - Use concise, descriptive parameter names aligned with the **Parameters and Types** of the current sub-goal.
  - Ground each step in actual primitive actions: `go to`, `open`, `take`, `put`, `move`, `clean`, `heat`, `cool`, `use`.
  - Express branches and loops using production rules in the form: "If <condition>, then <primitive action>". You may use "Else if" and a final default action when appropriate.
  - Represent loops compactly as: "For each <X> in <LIST>: <primitive action or rule>" (keep total steps $\leq$ 5).
  - Express condition in natural language with the parameters declared in "parameters".
  - Minimum steps policy: ensure the "steps" array contains 2-5 items and includes at least two non-redundant primitive actions across all rules; do not output single-step sub-goals.

- **Applicable Tasks**: For each sub-goal, provide tasks where it applies.

  - Copy task names verbatim from the source data (no paraphrasing, no reformatting, preserve original casing and punctuation).

# Output Format:

```json {{
[
    {{
    "name": <sub_goal_1_name, ...>,
    "parameters": [
        "param_name_1: param_type_1",
        ...
    ],
    "start_conditions": [ ... ],
    "success_conditions": [ ... ],
    "description": "<sub-goal 1 description>"
    "steps": [ ... ],
    "applicable_tasks": [ ... ],
    }}, ...
]
}}
```

## SKILL INDUCTION PROMPT

You are a workflow synthesizer.
Your task:

- Read the sub-goal and the interaction trajectories, then instantiate the scaffold above to solve ONLY that sub-goal.

- For each sub-goal, output ONE Mermaid flowchart style (node-bound inputs, edge-only control). No prose.

# Sub-goal to implement

{sub_goal}
**# Data trajectories**
Here are some interaction trajectories:
{examples}
**Overall Guideline (must follow exactly):**

- Use lowercase names for local variables and UPPERCASE names for globals (e.g., `target_rec` vs `TARGET_RECEPTACLES`).

- Local and global identifiers must stay distinct; do not reuse the same base name with different casing.

- Use double braces {{VAR}} for all placeholders evaluated at run time.

- All effects on global state happen inside `DataOp/LoopControl` nodes via `writes GLOBAL: (...)`. `Interface/Check/Action` MUST NOT include `writes GLOBAL`.

- Branching is ONLY via `Check` nodes with labels Yes/No. `LoopControl` uses `body/done`.

- Edge mid-labels can be comma-separated.

- For any edge that enters a `LoopControl` node: use `Start_Loop` to indicate entering/resetting from outside (restart enumeration), and `Continue_Loop` to indicate continuing the current loop.

- Every `Check/Action` with parameters must make them resolvable from GLOBALS or inline bindings in `local in: (...)`.

- Respect data-flow ordering: any GLOBAL referenced in a node's `local in` must already be assigned by an earlier node via `writes GLOBAL`.

- Avoid unescaped quote characters within quoted strings; escape them to keep JSON and Python strings valid.

- Selection helpers `select_one` and `select_all` can be used only inside `DataOp` nodes.

- Node IDs define unique instances. When the same type of node is needed multiple times, duplicate the node by assigning unique IDs (e.g., `A_OPEN_1`, `A_OPEN_2`).

**Domain predicates (allowed inside DataOp/Check nodes):**

- `locates({{receptacle_name}}); reachable({{receptacle_name}})`

- `contains({{receptacle_name}}, {{item_name}}); holding({{item_name}})`

- `is_open({{receptacle_name}}); is_closed({{receptacle_name}})`

- `is_cleaned({{item_name}}); is_theated({{item_name}});`
  `is_cooled({{item_name}}); is_turned_on({{item_name}})`

- type checks: `is_item_of_type({{item_name}}, {{target_type}});`
  `is_receptacle_of_type({{receptacle_name}}, {{target_type}})`

- Quantifiers, e.g. `exists("Item", lambda x: contains({{receptacle_name}}, x)`
  `and is_item_of_type(x, {{target_type}}))`

**Object Selection (ONLY allowed inside DataOp nodes):**

- `select_one('Item' or 'Receptacle', FOL_expression)`

- `select_all('Item' or 'Receptacle', FOL_expression)`

**Reference Mermaid Flow (example for a different sub-goal)** Note: The example below targets a different sub-goal and is only to illustrate structure
and conventions. Do not copy its semantics if they do not match the current sub-goal. {refered_mermaid}
**Strict style scaffold (copy-paste exactly, then instantiate only the nodes you need):**

```
%%{{init: {{'theme': 'default',
  'themeVariables': {{'background': '#ffffff'}} }} }}%%
flowchart TD
    %% ===================== Class Definitions =====================
    %% Guideline: Strictly follow the class definitions.
    classDef Spec fill:#f4f4ff,stroke:#6a6ab2,stroke-dasharray: 4 3,
        stroke-width:2px;
    classDef Interface fill:#e2e2f2,stroke:#6a6ab2,stroke-width:2px;
    classDef LoopControl fill:#f9e4b7,stroke:#b99b37,stroke-width:2px;
    classDef PrimitiveAction fill:#f9c2c2,stroke:#c23737,stroke-width:2px;
    classDef Check fill:#d0e1f9,stroke:#4378a2,stroke-width:2px;
    classDef DataOp fill:#f0f0f0,stroke:#888888,stroke-width:2px;

    %% ===================== Type Alias Legend =====================
    %% Guideline: Strictly follow the type alias legend.
    LEGEND["Type Legend:
ReceptacleName := str
ItemName := str
        
ObjectType := str
ReceptacleType := str
Bool := bool
        
List_T := sequence of T
Optional_T := T or None"]:::Spec

    %% ===================== Spec =====================
    %% Guideline: Strictly follow the Spec.
{FLOW_SPEC_NODE}

    %% ===================== Interface & Spec =====================
    %% Guideline: START is entry; *_END are exits. No computations here.
    %% Guideline: Strictly follow the interface definition.
{START_INTERFACE_NODE}
    SUCCESS_END(["Interface: SUCCESS_FLAG: Bool:=True
        writes GLOBAL: (SUCCESS_FLAG: Bool:=True)"]):::Interface
    FAILURE_END(["Interface: SUCCESS_FLAG: Bool:=False
        writes GLOBAL: (SUCCESS_FLAG: Bool:=False)"]):::Interface

    %% ===================== Primitive Actions =====================
{ACTION_NODE}
    %% Guideline: Select the PrimitiveActions Nodes that workflow needs.
    A_GOTO["PrimitiveAction: (action: 'go to {{goto_rec}}')
        local in: (goto_rec: ReceptacleName = {{CURRENT_RECEPTACLE}})
        out: (executed: Bool)"]:::PrimitiveAction
    A_OPEN["PrimitiveAction: (action: 'open {{open_rec}}')
        local in: (open_rec: ReceptacleName = {{CURRENT_RECEPTACLE}})
        out: (executed: Bool)"]:::PrimitiveAction
    A_TAKE["PrimitiveAction: (action: 'take {{take_item}} from {{from_rec}}')
        
local in: (take_item: ItemName = {{ITEM_TO_TAKE}},
        from_rec: ReceptacleName = {{CURRENT_RECEPTACLE}})
        out: (executed: Bool)"]:::PrimitiveAction
    A_MOVE["PrimitiveAction: (action: 'move {{move_item}} to {{to_rec}}')
        local in: (move_item: ItemName = {{ITEM_TO_MOVE}},
        to_rec: ReceptacleName = {{CURRENT_RECEPTACLE}})
```

```
              out: (executed: Bool)"]:::PrimitiveAction
    A_CLEAN["PrimitiveAction: (action: 'clean {{clean_item}} with {{clean_rec}}')
        
local in: (clean_item: ItemName = {{ITEM_TO_CLEAN}},
        clean_rec: ReceptacleName = {{CURRENT_RECEPTACLE}})
        out: (executed: Bool)"]:::PrimitiveAction
    A_HEAT["PrimitiveAction: (action: 'heat {{heat_item}} with {{heat_rec}}')
        
local in: (heat_item: ItemName = {{ITEM_TO_HEAT}},
        heat_rec: ReceptacleName = {{CURRENT_RECEPTACLE}})
        out: (executed: Bool)"]:::PrimitiveAction
    A_COOL["PrimitiveAction: (action: 'cool {{cool_item}} with {{cool_rec}}')
        
local in: (cool_item: ItemName = {{ITEM_TO_COOL}},
        cool_rec: ReceptacleName = {{CURRENT_RECEPTACLE}})
        out: (executed: Bool)"]:::PrimitiveAction
    A_USE["PrimitiveAction: (action: 'use {{use_item}}')
        local in: (use_item: ItemName = {{DESKLAMP_TO_USE}})
        out: (executed: Bool)"]:::PrimitiveAction

    %% ===================== LoopControl (use only if needed) ==============
    %% Guideline: Only foreach loops via LoopControl nodes.
    LOOP_FOR_LOCATIONS["LoopControl: For receptacle_i in {{loop_receptacles}}
        
writes GLOBAL: (CURRENT_RECEPTACLE: ReceptacleName:=receptacle_i)
        
local in: (loop_receptacles: List_ReceptacleName =
        {{RECEPTACLE_CANDIDATES}})"]:::LoopControl
{LOOP_FOR_NODE}

    %% ===================== Checks =====================
    %% Guideline: Checks branch on Yes/No only.
    C_IS_CLOSED["Check: is_closed({{rec_name_to_check}})
        local in: (rec_name_to_check: ReceptacleName =
        {{CURRENT_RECEPTACLE}})"]:::Check
    C_HAS_TYPE["Check: exists(Item, lambda x: contains({{rec_name_to_check}},
        x) and is_item_of_type(x, {{item_type_to_check}}))
        local in: (rec_name_to_check: ReceptacleName =
        {{CURRENT_RECEPTACLE}}, item_type_to_check: ItemType =
        {{TARGET_ITEM_TYPE}})"]:::Check
{CHECK_NODE}

    %% ===================== Data Operation =====================
    %% Guideline: All GLOBAL writes and data-binding happen in DataOp nodes.
{D_INIT_NODE}
    D_SELECT_ONE_ITEM_WITH_TYPE["DataOp: writes GLOBAL: (TARGET_ITEM:
        ItemName = select_one(\'Item\', is_item_of_type(x,
        {{select_item_type}})))  
local in: (select_item_type: ItemType
        = {{TARGET_ITEM_TYPE}})"]:::DataOp
    D_SELECT_ALL_RECEPTACLE_WITH_TYPE["DataOp: writes GLOBAL:
        (SELECTED_RECEPTACLES: List_ReceptacleName = select_all(\'Receptacle\',
        is_receptacle_of_type(x, {{select_rec_type}})))  
        local in: (select_rec_type: ReceptacleTypeName =
        {{TARGET_REC_TYPE}})"]:::DataOp
```

```
{DATA_OP_NODE}

    %% ==================== Node Class Assignments ====================
{CLASS_ASSIGNMENTS}

    %% ==================== Legend links ====================
    START -.- FLOW_SPEC
    START -.- LEGEND

    %% ==================== Control-Flow Edges ====================
{CONTROL_FLOW_EDGE}

{HINTS}
```

## SKILL-ACTION ROUTER PROMPT

You are a skill/action router for an embodied household agent. Decide which skill or raw action should be executed next.
**# Task and history**
{task_message}
**# Current world facts (set semantics)**
{facts_current}
**# Task guideline**
{task_guideline}
**Routing steps:**

1. Pick the active or next sub-task from the `ordered_subtasks` in the task guideline using the history and facts.

2. For that sub-task, compare triggers/pre/post contexts in `skill_guidelines` and `raw_action_guidelines` against the current facts/intents.

3. Propose up to 3 candidates (skills preferred) with evidence and expected outcome.

4. Recommend one candidate; if none apply, return type "none" and list missing info.

**Strict JSON ONLY (no prose, no code fences):**

```
{
  "active_subtask": "<sub-task name from ordered_subtasks>",
  "subtask_reason": "<why this sub-task is active/next>",
  "candidates": [
    {
      "type": "skill|action",
      "name": "<skill name or raw action name>",
      "phase": "<phase/sub-task it aligns to>",
      "trigger_match": ["<triggers or pre-context cues that match>"],
      "supporting_facts": ["<facts that justify applicability>"],
      "expected_outcome": "<state change or follow-up intent>",
      "confidence": 0.0
    }
  ],
  "recommendation": {
    "type": "skill|action|none",
    "name": "<recommended name or none>",
```

```
    "why": "<short rationale>"
  },
  "missing_info": ["<what evidence is needed if nothing applies>"]
}
```

---

**SKILL-ACTION EXECUTION PROMPT**

Interact with a household to execute the routed skill/action.
# **Routing recommendation (follow unless impossible)**
{routing_recommendation}
# **Here are two examples:**
{examples}
# **Here is the task.**
{task_message}
# **Current world facts:**
{facts_current}
# **Available actions:**

- `think`: ¡thought about the overall task goal, sub-goals decomposition, current state, and the next sub-goal¿

- `go to`: ¡ReceptacleName¿

- `open`: ¡ReceptacleName¿

- `take`: ¡ItemName¿ from ¡ReceptacleName¿

- `move`: ¡ItemName¿ to ¡ReceptacleName¿

- `clean`: ¡ItemName¿ with ¡ReceptacleName¿

- `heat`: ¡ItemName¿ with ¡ReceptacleName¿

- `cool`: ¡ItemName¿ with ¡ReceptacleName¿

- `use`: ¡ItemName¿

# **Available skills:**
{skill_info}
# **Skill/Action Selection Guideline:**
{task_guideline}
{action_check_error}

- Honor the routing recommendation. If `recommendation.type` is "skill", set `skill` to that name and `action` to null. If `recommendation.type` is "action", set `action` to the raw action string and `skill` to null. Only deviate if impossible, and explain in reasoning.

- Do not repeat the same action/skill if nothing happened or failed.

- **Prioritize using the provided skills**.

- For the selected skills, follow the **Guidelines for your Parameter Bindings** to ensure the skill is called correctly to achieve the intended effect.

- If none of the skills are applicable, **then** select the action from the available actions.

Please enter your action in a json format.

```json
{
    "reasoning": <thought on action selection based on overall goals,
                  current states, and sub-goals; note if deviating
                  from recommendation>,
    "skill": <The selected skill. Return null if none of the skills
              are applicable or if a raw action is recommended.>,
    "start_condition_evidence": <thought on the start conditions
                                 of the selected skill>,
    "pre_span_summary": <summary in natural language relevant events
                         that occurred **before** the current
                         skill/action begins...>,
    "intent": <the intent of the current skill/action, must stay
               self-contained>,
    "parameter_bindings": {
      <skill-param-1> : <value-1>,
      ...
    },
    "action": <The selected action if using a raw action. If a skill
               is selected, set this field to null.>
}
```

