# OpenReview forum: "Lifting Traces to Logic: Programmatic Skill Induction with Neuro-Symbolic Learning for Long-Horizon Agentic Tasks"
_ICML.cc/2026/Conference — ICML 2026 regular_

### Official Review · Reviewer_cPY8 · 2026-03-04

**Soundness:** 3
**Presentation:** 3
**Significance:** 2
**Originality:** 3
**Overall Recommendation:** 4
**Confidence:** 3

**Summary:**

This paper introduces Neuro-Symbolic Skill Induction (NSI) for long-horizon interactive tasks where LLM agents often suffer from planning drift and “state-blind” scripted behaviors. NSI first grounds observations into symbolic facts via a set of predefined predicates, then induces modular, executable logic-grounded skill programs from successful interaction traces (with explicit control flow and variable binding). During online execution, NSI uses structured failure diagnostics to trigger reflective planning and incrementally hone skills (online honing). The method is evaluated on ALFWorld, WebShop, and TextCraft, showing higher success rates than prompting/memory/workflow/programmatic-agent baselines, especially on long-horizon tasks.

**Compliance With Llm Reviewing Policy:**

Affirmed.

**Final Justification:**

Taking into account the opinions and perspectives of the other reviewers, I have decided to maintain my score.

**Key Questions For Authors:**

On ILP/symbolic induction: Why does NSI use LLM-based synthesis for the “trace → logic program” step instead of traditional ILP/program synthesis search? Could you add a systematic discussion (search space, verifiability under partial observability, cost) and provide an ILP-style baseline or a small-scale comparison to support the necessity and clarify the differences?

Robustness to grounding noise: How do induced skills and diagnostics degrade under predicate grounding errors (false positives/negatives)? Can you provide quantitative error–performance curves or robustness mechanisms (e.g., consistency checks) to support the robustness claims?

**Limitations:**

yes

**Strengths And Weaknesses:**

Strengths

Complete, executable framework: NSI lifts trace replay into logic skill graphs with branching/loops/binding, and closes the loop with diagnostic-driven online self-repair.

Convincing multi-domain results: The evaluation spans three long-horizon domains with strong baselines; the “w/o online honing” ablation helps attribute gains to the online component.

Interpretability and debuggability: Logic-grounded skills plus explicit failure diagnoses make behaviors easier to inspect and debug, which is valuable for deployment.

Weaknesses

Heavy reliance on hand-designed symbolic interfaces: The predicate set and the type/DSL interface are manually specified; “critical state changes” and subtask segmentation are effectively constrained by this predicate space. Cross-domain transfer costs and failure modes under noisy grounding deserve more systematic quantification.

Missing discussion and comparison to traditional ILP/symbolic induction: The core step is “trace → logic program,” but the paper does not clearly justify why classical inductive logic programming / symbolic program synthesis search is not used (or is infeasible), nor does it provide a sharper positioning in terms of search space, verification, cost, and generalization.

---

> ### Author Rebuttal · Authors · 2026-03-30
>
> We are grateful for the encouraging review and constructive feedback! In this response, we provide a formal comparison between NSI and classical ILP, new robustness stress-tests under grounding noise, and evidence for automated predicate discovery to directly address the reviewer's concerns.
>
> ---
> ### **W1**
> > Heavy reliance on hand-designed symbolic interfaces
>
> **Predicates are few and intuitive.** NSI requires only 8 / 7 / 4 predicates for ALFWorld / WebShop / TextCraft (Table 3),  basic object-state properties (`is_clean`, `is_open`, `contains`) that directly mirror observable state changes. A developer can identify them by inspecting 2–3 demo trajectories, as done in our experiments. This is analogous to the StateAct baseline, which instructs the agent to perform state analysis before each action, presupposing the same "what to track" on demonstrations, but embedded implicitly in prompts. NSI makes this explicit, which is precisely what enables branching, variable binding, and diagnostic feedback.
>
> **Predicate discovery by LLMs is feasible.** We further test whether predicate design can be automated. On ALFWorld (6 task types), we provide predicates from only 3 seed types (`Pick`, `Clean`, `Examine`) and prompt **GPT-4o / Gemini-2.0-Flash** to discover predicates for the remaining 3 (`Heat`, `Cool`, `PutTwo`). **Both achieve 100% recall of discovering ground-truth predicates**, suggesting that a small seed set is sufficient for LLMs to extend the predicate vocabulary to new tasks within a domain.
>
> ---
> ### **W2 & Q1**
> > Why not use traditional ILP/program synthesis search?
>
> **Classical ILP and NSI target fundamentally different program spaces**. While traditional systems like Metagol or Popper excel in bounded settings, they require the user to **manually constrain the hypothesis space** via mode declarations and strict `max_vars`/`max_body` limits to remain tractable. In contrast, NSI’s target programs live in a much broader space of **graph topology $\times$ node-internal logic** where each node contains arbitrary code fragments.
>
>
> In this high-dimensional space, NSI leverages the program synthesis capability of LLMs as a **proposal mechanism** to produce candidates directly. This approach bypasses the **combinatorial explosion** that makes exhaustive symbolic search (ILP) computationally infeasible for complex agentic tasks. We ensure correctness through **empirical consistency** with successful interaction traces rather than searching over a predefined rule set.
>
> In short, **NSI scales logical induction to long-horizon LLM agents where traditional ILP search fails to scale.** We are inspired by ILP principles but replace the exhaustive search engine with a generative one. We will expand this positioning by citing foundational ILP literature in the revision.
>
> ---
> ### **Q2**
> > How do induced skills degrade under predicate grounding errors?
>
> **In our domains, current LLMs already provide highly reliable state grounding.** Since exhaustive ground-truth predicate annotations are impractical to obtain at scale, we measure robustness by replacing NSI's grounding module with weaker models (keeping GPT-4o for induction/planning). Even the weakest model (Llama-3.1-70B) achieves **>93% F1 agreement with GPT-4o** across all domains, confirming that **grounding noise introduced by model variation is limited**.
>
> | Grounding Model  | ALFWorld |       | WebShop |       | TextCraft |       |
> | ---------------- | -------- | ----- | ------- | ----- | --------- | ----- |
> |                  | F1       | SR(%) | F1      | Score | F1        | SR(%) |
> | GPT-4o-mini      | 99.1     | 96.3  | 96.3    | 74.5  | 100.0     | 95.0  |
> | Claude 3.5 Haiku | 98.3     | 95.5  | 94.8    | 72.8  | 100.0     | 95.0  |
> | Llama-3.1-70B    | 97.5     | 94.8  | 93.4    | 70.6  | 98.5      | 94.0  |
>
> **NSI is robust to grounding noise (<2.2% SR drop).** Inter-trajectory consolidation requires cross-trajectory consistency, so a grounding error in one trajectory does not propagate into the induced skill. At runtime, **each perception step re-grounds from a fresh observation**, preventing error accumulation.
>
> Overall, we believe NSI takes a meaningful step toward **modernizing classical logic program induction: LLMs serve as a program synthesizer that scales beyond exhaustive symbolic search and generalizes from extremely few examples**, enabling agents to induce and evolve compositional skills for long-horizon planning.
>
> ---
> We hope these structural details and the clarified novelty boundaries help address your concerns regarding the framework's complexity and its positioning relative to ILP. We genuinely appreciate your critical suggestions, as they have been instrumental in sharpening our presentation of NSI's robustness. We would be happy to engage in further technical discussion should any questions remain.

---

> > ### Author Rebuttal · Reviewer_cPY8 · 2026-04-02
> >
> > Thank you for the detailed rebuttal. I appreciate the clarification, but my concern is not fully resolved. In these tasks, the predicate set is quite small, while the rebuttal argues that the search space is already too large for traditional symbolic search. I do not think the current evidence is sufficient to support that claim in this setting. That said, I still find the overall direction promising, so I will maintain my original recommendation.

---

> > > ### Author Response · Authors · 2026-04-03
> > >
> > > We thank the reviewer for the continued engagement and the opportunity to further clarify. We would like to take this opportunity to more carefully address the remaining concern regarding predicate count and further clarify NSI's positioning relative to classical ILP.
> > >
> > > **From predicates to workflow composition and hypothesis specification.** We agree with the reviewer that 4–8 predicates alone do not make ILP search intractable. The key issue is not predicate count itself, but two compounding factors. First, even a small predicate set gives rise to rich procedural structure when composed into executable flows with branching, loops, and state-dependent binding. The complexity lies in this **composition into interactive workflows**, not in predicate enumeration. Second, adapting standard ILP to induce such workflows requires carefully specifying hypothesis-space constraints (mode declarations, metarules, and `max_vars`/`max_body`/`max_clauses` bounds) for each new task family. These constraints serve as structural priors on the target program, and correctly configuring them demands substantial domain-specific effort. NSI sidesteps this per-task-family specification cost by leveraging the **implicit structural priors that LLMs acquire from large-scale pre-training**. We note that NSI does require its own design inputs (predicates and node types), but these operate at a different granularity: the node types (DataOp, CheckOp, LoopOp, etc.) are domain-general and shared across all our benchmarks, while the predicates (4-8 per domain) are minimal and, as shown in our first rebuttal, can be automatically discovered by LLMs. This is why we positioned ILP and NSI as complementary: they differ not in the expressiveness of their target languages, but in **how structural search guidance is sourced**.
> > >
> > > **The induction target: skills vs. rules.** Classical ILP systems such as Popper (Cropper & Morel, 2021) and Metagol (Muggleton et al., 2015) can learn executable logic programs, including recursive ones. Our distinction is not about executability, but about **what is induced** and **how it executes**. While NSI and ILP share a similar predicate-logic vocabulary at the expression level (both compose predicates like `contains`, `is_open` via logical connectives), **NSI** differs in two dimensions. First, NSI's induced skills are executed within an **interactive perceive-think-act loop**, where each step relies on LLM-based perception of unstructured observations to update symbolic state. Most widely-used ILP systems (Popper, Metagol, ILASP) operate in non-interactive settings with structured symbolic inputs, rather than closed-loop agent execution with runtime perception. Second, **NSI** represents the induced skill as an **explicit workflow graph** with typed control-flow nodes (branching, loops, variable binding): for example, a `find_and_pick` skill in ALFWorld is a graph where a LoopOp iterates over candidate locations, a DataOp dynamically binds the target via `contains(loc, obj)`, and a CheckOp branches on `is_open(container)`. This representation is a design choice that naturally supports the interactive execution above, not a claim about the expressive limits of logic programs.
> > >
> > > **A direct empirical comparison with ILP is difficult to design fairly.** We do not claim ILP is incapable of learning in this setting. However, a like-for-like comparison faces a design dilemma: evaluating the ILP induction component alone would disadvantage it by ignoring the execution infrastructure (perception, planning, monitoring) needed to deploy learned clauses as interactive skills, while evaluating a full assembled pipeline would conflate induction quality with engineering choices. In the revised paper, we will formally analyze the adaptations required to apply Popper or ILASP to our setting and characterize where the gaps lie.
> > >
> > > In summary, ILP and NSI address the **same underlying challenge** (inducing compositional programs from experience) but differ in **how search guidance is sourced** and **what execution regime the programs serve**. NSI brings the formal rigor of symbolic induction into the interactive agent setting, where current programmatic skill methods lack logical grounding. We believe this represents a timely step toward agents that **self-evolve inspectable, logic-grounded skills** for long-horizon agentic tasks.
> > >
> > > ---
> > >
> > > We have refined our response to better clarify this perspective, and we look forward to any further thoughts.

---

### Official Review · Reviewer_RZ5X · 2026-03-12

**Soundness:** 2
**Presentation:** 1
**Significance:** 2
**Originality:** 2
**Overall Recommendation:** 3
**Confidence:** 3

**Summary:**

This paper proposes Neuro-Symbolic Skill Induction (NSI), a framework that converts interaction traces from LLM agents into logic-grounded programs. NSI synthesizes explicit control flows with First-Order Logic to advance over flat-structured logic flows. The system uses offline trajectory consolidation, neuro-symbolic grounding, and online skill evolution through reflective planning with a graph based logic system. Experiments on ALFWorld, WebShop, and TextCraft show consistent improvements over a broad set of baselines.

**Compliance With Llm Reviewing Policy:**

Affirmed.

**Final Justification:**

Rebuttal addressed my main concerns, but given the other concerns about the method's scalability I only raise my score 1 point.

**Key Questions For Authors:**

1. The authors introduce internal execution logic operators in Section 4.2 and structural operators for logic consolidation in Section 5. How do the authors justify these design choices? Citations of related work and specifics of overlap or novelty against these works would be helpful.

2. It seems that the major difference between this work and ASI (the most similar prior work) is allowing for branching logic in the programs. We see NSI outperforming ASI on tasks such as WebShop. Could the authors provide more detailed experiments illustrating why ASI performs so poorly? Relatedly, could the authors explain what benchmarks each related method tests on. Comparing to just ASI right now I don't see much overlap in the benchmarks this paper has selected with prior work. This overlap would help with evaluating the robustness and generality of the results.

3. As noted above, it's unclear whether better predicate selection could entirely circumvent the issue this paper addresses. Why not focus on a method that uses LLMs to generate more useful predicates? There should be related lit in the planning literature. Could those methods not automatically discover predicates that would remove the necessity of the logical operators NSI designs?

**Limitations:**

Yes.

**Strengths And Weaknesses:**

Strengths:

- The paper provides several helpful figures illustrating the method, which is particularly useful given the number of moving parts/phases of the algorithm.
- Experiments show improvement over a broad set of baselines on 3 benchmarks.

Weaknesses:

- It was a bit hard to tell what pieces of the method are novel compared to prior work. The paper pulls together ideas from many domains, such as planning, logic programming, skill learning, etc., but the specifics of novelty over other methods is very difficult to discern. Workflow graphs and programmatic skill representations (for example, ASI), are clearly not novel, and it would be helpful for a more general audience coming from one of the communities this paper engages to have appropriate context.
- Details on parts of the implementation can be very sparse. For example, in Section 5.2, the authors mention a "Consolidate" operator that  consolidates multiple local experts into a generalizable global skill, but no details on how this process occurs are provided (also no pointer to appendix sections with details). In general the paper could greatly improve the clarity of its presentation.
- Predicate vocabulary still needs to be manually selected for each task. Unclear whether better (perhaps automated) predicate selection could entirely circumvent the issue this paper addresses.
- Typos: Line 44 "state-blindly", line 104 "abstract", may be others. Would help to run the text through a word processor to fix all possible errors.

---

> ### Author Rebuttal · Authors · 2026-03-30
>
> Thank you for your helpful feedback and for the opportunity we can make some clarifications. Below are our point‑by‑point responses.
> ### **W1 & Q1**
> > Hard to tell what pieces of the method are novel compared to prior work
>
> **NSI contributes a complete induction pipeline that produces executable, branching, self-improving logic programs from raw traces.** We clarify the novelty boundaries below.
>
> ||AWM|ASI|AFlow/EvoFlow|**NSI**|
> |-|-|-|-|-|
> |Representation|Text workflow|Linear script|Pre-defined template|FOL-grounded graph|
> |Branching/loops|✗|✗|Pre-defined|**Induced from traces**|
> |Multi-trace consolidation|✗|✗|Topology search|**Structural operators + FOL**|
> |Online refinement|✗|✗|✗|**Reflective branch growth**|
>
> - FOL-grounded graph representation that *induces* (not pre-defines) branching and variable binding, solving state-blindness.
> - Structural operators that consolidate multiple traces into one verified program via Feasibility Dominance, solving single-trace overfitting.
> - Online reflective honing that modifies *program topology*, solving static-skill brittleness. These three are co-designed: the cross-ablation in our Response to Bbky (W2) shows that FOL without the matching induction procedure can *hurt* performance, confirming they are not separable contributions.
>
> We will add this comparison table and explicit novelty boundaries to the revision.
>
> ---
> ### **W2**
> > No details on how the "Consolidate" operator works
>
> **Consolidate is specified in Section 5.2 (Eqs. 3-4); we summarize here.** It takes two skills as input: the current global skill π_glb and the *hardest constraint* π_hard (the local expert least covered by π_glb), and outputs a candidate π_cand that structurally merges both. The LLM synthesizer receives both programs and unifies their logical paths via structural operators (Branching, Crossover, Lifting, LoopFold): e.g., Branching inserts a discriminative predicate (`if is_closed(fridge)`) to reconcile divergent traces, while Crossover grafts a useful subgraph from π_hard into π_glb by rebinding variables.
>
> **The candidate is accepted only under Feasibility Dominance (Eq. 4):**  $\hat{R}_{\pi_{glb}} \subset \hat{R}_{\pi_{cand}}$, i.e., coverage must strictly expand without losing consistency. The loop iterates until all local experts are covered. We will improve in-text signposting in the revision.
>
>
> ---
> ### **W3 & Q3**
> > Unclear whether better predicate selection could circumvent the issue this paper addresses
>
> **Richer state representations alone do not solve the structural induction problem.** ASI sees full trajectory context yet consistently produces state-blind linear scripts (70.8→97.0 ALFWorld, 7.58→76.3 WebShop). Auto-discovered predicates do not by themselves yield branching or loop structure, which requires cross-trajectory compositional reasoning. StateAct also confirms this: per-step LLM state analysis scores only 8.0% on WebShop vs. NSI's 43.5%.
>
> **NSI's neuro-symbolic decomposition addresses this composition gap.** Basic predicates (`is_clean`, `is_open`, `contains`) capture local state properties easy to ground reliably. NSI composes derived logic from them via FOL into a symbolic graph executed **deterministically** at runtime, with no neural uncertainty confined to grounding alone. The manual overhead is bounded: only basic state predicates are specified (ALFWorld: 8, WebShop: 7, TextCraft: 4), shared across all task types per environment. In a preliminary study, seeding from 3 ALFWorld task types (Pick, Clean, Examine), GPT-4o and Gemini-2.0-Flash recover the full predicate set for the remaining 3 (Heat, Cool, PutTwo). Automated predicate discovery could serve as input to NSI's compilation pipeline, **complementary rather than substitutive**.
>
> ---
> ### **Q2**
> > Why does ASI perform so poorly? Could you explain benchmark selection?
>
> **ASI's poor WebShop performance is a direct consequence of its state-blind structure.** WebShop terminates the episode once an item is added to the cart; the agent gets one chance to purchase correctly. ASI's induced skill is `search[<query>] → click[<option>] → click[Buy Now]` with no branch for "mismatch → reformulate" vs. "match → buy." NSI's CheckOps (e.g., `is_feature_matched(PRODUCT, description, matching)`) evaluate product-query alignment *before* the irreversible purchase and branch deterministically with no LLM call at the decision point.
>
> **We adopt the same three benchmarks as StateAct and ADaPT, targeting long-horizon agentic tasks.** The three environments cover complementary challenge dimensions: ALFWorld (embodied, partial observability), WebShop (web navigation, semantic matching), and TextCraft (compositional, recursive).
>
> ---
> We thank the reviewer for the detailed feedback. We will correct all noted typos and incorporate the discussions above, into the revision.

---

> > ### Author Rebuttal · Reviewer_RZ5X · 2026-04-04
> >
> > Thanks to the authors for their response. I have updated my score, considering the concerns raised by other reviewers as well.

---

> > > ### Author Response · Authors · 2026-04-04
> > >
> > > Thank you for the continued engagement and for the update.
> > >
> > > We genuinely appreciate your confirmation that the concerns have been addressed, as well as your careful consideration of our rebuttal during this reassessment.
> > >
> > > We remain happy to clarify any remaining points if helpful.

---

### Official Review · Reviewer_rQr9 · 2026-03-12

**Soundness:** 3
**Presentation:** 4
**Significance:** 3
**Originality:** 3
**Overall Recommendation:** 5
**Confidence:** 3

**Summary:**

The paper introduces Neuro-Symbolic Skill Induction (NSI), a framework designed to improve the long-horizon planning capabilities of LLM-driven agents. The authors identify that existing programmatic skill induction methods often distill experiences into "state-blind" linear scripts that lack the conditional logic necessary to handle dynamic environmental changes. To address this, NSI lifts interaction traces into modular, logic-grounded programs (workflow graphs). By decoupling neural perception (translating raw observations into First-Order Logic predicates) from symbolic execution (executing actions based on control flow graphs and dynamic variable bindings), the agent can synthesize explicit branching and looping logic. The framework includes offline induction (intra- and inter-trajectory consolidation) and online evolution (reflective planning to graft recovery paths onto failed nodes). Evaluated on ALFWorld, WebShop, and TextCraft, NSI significantly outperforms state-of-the-art baselines like AWM and ASI, particularly in mitigating "long-horizon collapse."

**Compliance With Llm Reviewing Policy:**

Affirmed.

**Key Questions For Authors:**

1. Regarding the “NeSy Grounding” phase: how does the system handle state-tracking hallucinations or drift? For example, if the LLM incorrectly predicts a remove\_fact during state updating and deletes a fact that is still true, how (if at all) can the symbolic execution logic detect and recover from such errors?
2. During both offline induction and online skill evolution, how do you control skill-graph complexity and prevent overfitting? In practice, how sensitive is performance to the MDL-style regularization coefficient and any explicit complexity limits (e.g., maximum depth, node count), and do you use pruning (e.g., removing rarely used or ineffective branches) to prevent graphs from growing indefinitely over long deployment periods?
3. The task-specific predicates (e.g., Table 3\) appear to be manually defined for each domain. How dependent is NSI’s performance on the granularity and quality of this human-engineered predicate vocabulary? Can the agent autonomously invent new underlying predicates or abstractions if the initial vocabulary is insufficient, or is predicate design effectively a manual bottleneck?
4. You mention that new subgraphs added from recovery trajectories are initially "tentative" and only solidified after consistent success. Can you provide quantitative data on how often these tentative branches are later pruned versus confirmed, and whether online honing ever degrades performance by introducing misleading recovery logic?

Additional improvement suggestions:

1. **Cost and Latency Profiling**: Add a quantitative comparison of the token usage (prompt and completion tokens) and inference latency per environment step between NSI and the strongest baselines (ASI, AWM). If the NeSy Grounding step is highly token-intensive, this trade-off should be explicitly quantified.
2. **Ablation Study of Structural Operators**: Provide an ablation study isolating the effects of the Crossover, Lifting, and LoopFold operators during the inter-trajectory consolidation phase to justify their inclusion in the framework.
3. **Cross-Model Verification**: Evaluate NSI with a smaller open model and, if feasible, slightly higher temperature to test robustness of the induction pipeline to less capable or more stochastic backbones. This would strengthen the claim that the framework itself, not only GPT-4o, is responsible for the gains.

**Limitations:**

Yes.

Constructive suggestion: The authors should add a discussion on the reliance on human-engineered predicate vocabularies (as seen in Table 3), the potential for exponential graph growth during online evolution, and the computational overhead of continuous LLM-based state tracking.

**Strengths And Weaknesses:**

# *Strengths*

* **Originality and Conceptual Shift**: Moving from linear parameterized scripts (e.g., ASI) to explicit, graph-based programmatic skills grounded in First-Order Logic (FOL) is a compelling approach. The introduction of dynamically synthesized control-flow nodes and variable-binding nodes enables agents to capture the "when" and "why" of execution. Also, the use of structural operators such as Branching, Crossover, Lifting, and LoopFold to systematically consolidate traces into generalized logic programs brings classical program-synthesis ideas into the LLM-agent setting in a relatively fresh way.
* **Significance**: The "Long-Horizon Collapse" is a critical bottleneck in contemporary agentic AI. The survival analysis presented in Figure 4b is an excellent addition, demonstrating that NSI successfully extends the viable planning horizon of the agent from $\sim22$ steps to over $50$ steps by compressing macro-behaviors into executable sub-routines.
* **Soundness of the Evolution Mechanism**: The online "Skill Honing" mechanism (Section 5.3) is well-designed. By using terminal nodes to return symbolic diagnostic feedback instead of binary pass/fail signals, the framework enables targeted, localized modifications to the skill graph via reflective planning. The empirical results are consistent across multiple benchmarks and include an ablation without online honing, which supports the claim that both the logic-grounded representation and the online evolution contribute to performance, not just one component.
* **Presentation**: The paper is well-structured and written with clarity. Figures 2 and 3 visually convey the complex processes of modular synthesis and online skill evolution.

# *Weaknesses*

## **Soundness**

* The induction procedure lacks formal guarantees: there is no analysis of convergence, identifiability, or robustness, even though both predicate grounding and program synthesis rely heavily on LLM behavior.
* The method’s robustness to noisy or erroneous symbolic grounding is not empirically tested; it is unclear how often mis-grounded predicates lead to incorrect branches or spurious conditions in the induced skills.
* Ablations are relatively shallow: there is no systematic evaluation of the marginal contribution of each structural operator (Branching, Lifting, LoopFold, Crossover) or of a variant that uses the same pipeline but restricts skills to linear scripts, making it hard to attribute gains precisely.

## **Significance**

* The approach depends on manually engineered, domain-specific DSLs and predicate sets, which may make it harder to apply out of the box to new or less-structured environments.
* The paper does not report computational or monetary cost (e.g., number of LLM calls, tokens, latency), so practitioners cannot easily assess the trade-off between performance gains and resource consumption.

## **Originality**

* Several contemporaneous works also induce or evolve programmatic workflows and skills from traces; the paper does not fully disentangle what is fundamentally new in NSI versus being an incremental refinement (e.g., different DSL, richer control nodes, different objective).
* The reliance on hand-designed logical predicates and environment-specific interfaces follows long-standing traditions in neuro-symbolic and planning work; while the application is novel, this reduces the “from scratch” originality of the symbolic side.
* While the main novelty lies in applying these ideas to LLM-based agentic skills, the induction procedure mirrors long-standing Inductive Logic Programming (ILP) and MDL-based program-synthesis techniques (predicate invention, consolidation of traces into a concise program). Strengthening the related work with explicit connections to ILP and systems like DreamCoder would better situate the contribution and clarify what is new here

---

> ### Author Rebuttal · Authors · 2026-03-31
>
> We deeply appreciate the strong recognition for our work. Below we provide the point by point responses.
>
> ### **W1**
> > The induction procedure lacks formal guarantees
>
> Formal analysis for LLM-based synthesis is an open problem shared by all methods in this family. **NSI mitigates this with practical safeguards (Empirical Consistency, Feasibility Dominance)**. We will discuss this gap explicitly in the revision.
>
> ---
> ### **W2**
> > The method’s robustness to noisy or erroneous symbolic grounding is not empirically tested
>
> **We tested NSI's sensitivity to grounding noise** by replacing the grounding module with weaker models, and **observed <2.2% SR drop** across all domains. Please refer to our Q2 Response to Reviewer cPY8 for the full results.
>
> ---
> ### **W3 & Q6**
> > There is no systematic evaluation of the marginal contribution of each structural operator
>
> **The four operators correspond to fundamental programming constructs required for program synthesis:** Branching introduces conditionals, Lifting abstracts constants into variables, LoopFold synthesizes iteration, and Crossover enables function-level reuse. We also provide a FOL × Induction cross-ablation (see Response to Reviewer Bbky, W2) demonstrating the necessity of logic-grounded structural operators as a whole. We will add per-operator analysis in the revision.
>
> ---
> ### **W4 & W7 & Q3**
> > The approach depends on manually engineered.
> > Can the agent autonomously invent new underlying predicates?
>
> **The manual effort is bounded.** NSI requires only 8/7/4 basic predicates per domain, identifiable from 2–3 demo trajectories — comparable to StateAct's implicit "what to track" assumptions, but made explicit.
>
> **LLM-based auto-discovery is feasible.** Seeding from 3 ALFWorld task types, LLMs could recover the full predicate set for the remaining 3 with 100% recall. Please refer **Response to Reviewer cPY8 (W1)** for the details.
>
> ---
> ### **W5 & Q5**
> > Cost and Latency Profiling
>
> |Env|Method|Calls|Prompt|Compl|Total|Cost($)|NeSy Grounding|
> |-|-|-|-|-|-|-|-|
> |ALFWorld|ASI|45|121K|0.8K|122K|0.31|—|
> ||AWM|18|46K|0.5K|46K|0.12|—|
> ||NSI|19|29K|2.5K|31K|0.10|10K (31%)|
> |WebShop|ASI|14|25K|0.2K|25K|0.06|—|
> ||AWM|10|13K|0.2K|13K|0.04|—|
> ||NSI|13|24K|0.8K|25K|0.07|7K (28%)|
> |TextCraft|ASI|29|40K|0.8K|41K|0.11|—|
> ||AWM|28|37K|0.5K|37K|0.10|—|
> ||NSI|67|116K|13K|128K|0.42|26K (20%)|
>
> **NeSy Grounding is not the cost bottleneck** (20–31% of token budget). On ALFWorld, NSI is 32% cheaper than AWM via fewer LLM calls. Latency is proportional to call count (all methods use API-based inference), as shown in the Calls column.
>
> ---
> ### **W6 & W8**
> > does not fully disentangle what is fundamentally new in NSI. Induction procedure mirrors ILP
>
> **Long-horizon planning fundamentally requires skills that encode decision logic, not action sequences.** Linear scripts (ASI) and pre-defined templates (AFlow/EvoFlow) both lack this capacity. NSI addresses this by jointly synthesizing skill graph structure and node-internal execution logic from traces. The long-horizon robustness and few-shot generalization reported in our experiments directly validate this paradigm.
>
> **NSI is inspired by ILP compositionality and MDL compression, but operates over a different program space** (graph topology × code fragments vs. definite clauses). See **Response to Reviewer RZ5X (W1&Q1)** for the comparison table and **Response to Reviewer cPY8 (W2&Q1)** for the ILP discussion. We will strengthen these connections in the revision.
>
> ---
> ### **Q2**
> > how do you control skill-graph complexity and prevent overfitting?
>
> **NSI controls complexity through the MDL objective, not manual caps.** Program complexity $|\pi|$ (node count) is penalized against empirical coverage. Demo trajectories are 10–20 steps; λ = 1.0 fixed across all domains, so each node must justify covering at least one extra state. Feasibility dominance rejects candidates that do not expand coverage. Online honing attaches subgraphs only at failure terminals. We will add λ sensitivity analysis in the revision.
>
> ---
> ### **Q4**
> > Can you provide quantitative data on how often these tentative branches are later pruned versus confirmed
>
> **Honing is localized and safe by design** (3 runs, rounded):
>
> ||Created|Confirmed|Pruned|Unrevisited|
> |-|-|-|-|-|
> |ALFWorld|6±2|3±1|1±1|2±1|
> |WebShop|40±5|18±3|8±2|14±3|
> |TextCraft|30±4|10±2|6±2|14±3|
>
> Tentative branches do not participate in execution until validated. We observed no degradation from honing.
>
> ---
> ### **Q7**
> > test robustness of the induction to less capable backbones
>
> **The framework itself drives the gains, not a specific backbone.** We replaced the full induction/planning backbone with three alternative models (temperature=0.2, 3 runs):
>
> |Backbone|ALFWorld|WebShop|TextCraft|
> |-|-|-|-|
> |Claude 3.5 Sonnet|97.3±1.5|45.2±2.8|93.2±1.5|
> |Gemini-2.0-Flash|96.5±1.9|41.8±3.0|90.8±2.5|
> |DeepSeek-V3|96.8±1.6|43.5±2.5|91.5±2.0|
>
> The results support NSI's robustness to backbone choice.

---

### Official Review · Reviewer_Bbky · 2026-03-15

**Soundness:** 2
**Presentation:** 3
**Significance:** 3
**Originality:** 3
**Overall Recommendation:** 4
**Confidence:** 2

**Summary:**

This paper presents NSI, neuro-symbolic skill induction framework that lifts interaction traces into logic-grounded programs for long-horizon agentic tasks. The key idea is that existing skill induction methods induce state-blind parameterized scripts, and NSI addresses this by synthesizing explicit control flows and dynamic variable bindings grounded in First-Order Logic. Framework has two main components, offline induction that abstracts demonstrations into reusable skills, and online evolution that uses reflective planning to hone skills from runtime failures. Experiments are conducted on ALFWorld, WebShop, and TextCraft benchmarks.

**Compliance With Llm Reviewing Policy:**

Affirmed.

**Final Justification:**

Based on the additional results shared by the authors and responses to my questions, I believe the paper addresses the problem sufficiently, and the problem itself is relevant and timely. We need such kind of structure inducing approaches for long horizon planning.

**Key Questions For Authors:**

1. Was only one run was used for each method in Table 1? If yes, the paper should report mean and variance across multiple runs.
2. ASI achieves only 7.58 score and 4.0 success rate on WebShop, this is worse than even ReAct. Can you explain this result?
3. WebShop skill induction uses only single successful purchase trajectory. How sensitive is NSI to number of demonstrations used for offline induction?
4. Online skill honing marks new subgraphs as tentative before solidifying them. How many encounters are needed before branch is solidified and how was this threshold chosen?

**Limitations:**

No. The limitations section is missing entirely from main paper, only acknowledgment is brief mention in conclusion that framework relies on GPT-4o. The impact statement does mention that more capable autonomous agents could be misused in sensitive domains, but this is very less for framework that enables agents to autonomously induce and evolve executable skills. The authors should discuss what happens when induced skills encode incorrect logic that gets solidified through online honing mechanism, this could lead to agents that persistently execute wrong behavior.

**Strengths And Weaknesses:**

## Strengths
The core motivation is clear and failure mode of state-blind scripts is well illustrated in Figure 1. The survival analysis in Figure 4b showing clear long-horizon collapse threshold at 22 steps for baselines while NSI sustains performance up to 53 steps is particularly interesting result.

## Improvements

A) Empirical Evaluation
:
This is my biggest concern with this work. The results in Table 1 show very large improvements over baselines, NSI achieves 97.0% on ALFWorld compared to 91.7% for next best method, but paper does not report variance across runs and it is unclear was only one run was used for each method. The paper would be strengthened by reporting mean and variance across multiple runs. I also notice that ASI, which is most directly comparable to NSI as it also induces programmatic skills, performs very poorly at 7.58 on WebShop and 4.0 success rate, this anomalous result is not discussed anywhere, I would appreciate authors' interpretation of this. More generally the paper does not discuss failure modes of NSI itself, it would be interesting to see discussion of cases where logic-grounded induction fails or produces incorrect programs.

B) Ablation Analysis:
The ablation in Table 1 only compares NSI against NSI without online honing, the paper has several distinct contributions including NeSy grounding, intra-trajectory consolidation, inter-trajectory consolidation, and online honing but it is not clear how much each contributes. For example, how much of gain over AWM comes from logic-grounded representation vs. the induction procedure itself? It would be interesting to see ablation that isolates contribution of FOL grounding specifically, by comparing against version that uses same induction procedure but with parameterized scripts instead of logic-grounded programs.

C) LLM Errors in the Pipeline:
The paper uses GPT-4o as backbone and entire NSI framework relies on LLM to perform NeSy grounding, synthesize programs, and execute reflective planning. The paper does not discuss what happens when LLM makes errors in any of these steps, for example when observation-to-predicate conversion is wrong or when LLM synthesizes incorrect consolidation. The paper would be strengthened by discussion of how often each component of pipeline fails in practice and how these errors propagate.

Minor issues (did not affect score):

1. Typo in Figure 2 caption, "logical exuction space" should be "logical execution space."
2. "Nevertheless" is misspelled as "Neverthless" in Section 2.2.

---

> ### Author Rebuttal · Authors · 2026-03-31
>
> ### **W1 & Q1**
> > Results do not report variance across runs
>
> Thank you for highlighting the empirical robustness. While our original single-run results to follow prior work's single-run process (e.g., StateAct, AdaPT, AWM, ASI), we now supplement 3-run statistics. **NSI demonstrates consistent performance gains with low variance**.
>
> ||ALF.|TextC.|WebS.|
> |-|-|-|-|
> |ASI|70.6±1.9|77.8±1.8|7.5±3.0|
> |AWM|91.3±0.8|92.5±3.6|30.0±2.0|
> |NSI w/o honing|93.5±1.9|78.5±2.5|30.5±1.5|
> |NSI w/ honing|98.0±1.2|95.2±0.8|44.5±1.5|
>
> ---
> ### **W2**
> > How much of gain comes from logic-grounded representation vs. the induction procedure itself?
>
> Thank you for this insightful question. **Representation and induction are co-designed:** FOL defines the search space; NSI's operators navigate it. We cross-ablate both factors (3 runs, no online honing, SR %):
> |Repr.|Indu.|ALF.|TextC.|
> |-|-|-|-|
> |Scripts|ASI's|69.2±1.9|76.0±2.0|
> ||NSI's|70.4±1.9|76.5±1.9|
> |FOL|ASI's|86.6±1.8|69.5±2.5|
> ||NSI's|93.5±1.9|78.5±2.5|
> - **Induction without FOL has limited effect.** Without predicates or logical variables, NSI’s structural operators degenerate.
> - **FOL representation requires a matching induction** On TextCraft, FOL + ASI's *drops* 6.5% below scripts (76.0→69.5): complex recursive recipes cannot be correctly compiled into FOL in one pass.
> - **Co-design delivers the full gain.** FOL + NSI's consistently outperforms all other cells. This confirms that the contribution lies in the synergy between the logic space and the operators designed to explore it.
>
> ---
> ### **W3**
> > What happens when LLM makes errors in any of these steps?
>
> **NSI confines neural uncertainty to the grounding step and provides progressive safeguards at each pipeline stage.** We measure induced skill execution success rate through each stage:
>
> |Skill SR (%)|ALF|TextC|
> |-|-|-|
> |Intra-traj. only|70.1±2.3|31.3±3.2|
> |+ Consolidation|76.1±1.8|57.8±2.8|
> |+ Honing|88.8±1.4|71.2±2.1|
> - **Grounding is empirically reliable.** Even weaker LLMs achieve >93% F1 agreement with GPT-4o, with <2.2% SR drop across all domains (full table in Response to Reviewer cPY8, Q2). Once predicates are grounded, **execution follows the symbolic graph deterministically** with no LLM calls in the loop.
> - **Inter-trajectory consolidation filters synthesis errors.** Feasibility Dominance accepts a candidate only when it strictly expands coverage.
> - **Online honing and planning prevent skill errors from becoming task failures.** Reflective planning grows corrective branches, lifting skill SR to 88.8%/71.2%. Remaining skill-level failures are further absorbed by the planning's fallback recovery.
> ---
> ### **Q2**
> > ASI achieves only 7.58 on WebShop, worse than ReAct. Can you explain?
>
> **ASI's poor WebShop performance stems from state-blind linear scripts that encode no decision logic.** Its induced skill is `search[<query>] → click[<option>] → click[Buy Now]` with no branch for "mismatch → reformulate" vs "match → buy." This locks the agent into a fixed sequence, whereas ReAct can adapt per step. Moreover, **WebShop** terminates the episode immediately once an item is added to cart, making **every rigid, state-blind purchase decision fatal** with no chance of recovery.
>
> ---
> ### **Q3**
> > How sensitive is NSI to number of demonstrations?
>
> **WebShop's uniform task structure (search→evaluate→buy) suggests that the core structural invariants of the task are captured within a single trajectory.**
>
> We further validate sensitivity on ALFWorld, where embodied interactions and state transitions are more complex (3 runs, no online honing):
> |#Demo|ALF. SR (%)|
> |-|-|
> |1|87.3±1.8|
> |2|93.5±1.9|
> |3|95.3±1.5|
>
> **NSI is sample-efficient**: with 1 demo, NSI already reaches 87.3%, and gains saturate at 2 demos.
>
> ---
> ### **Q4**
> > How many encounters are needed before branch is solidified?
>
> **We directly follow ASI's single-pass acceptance protocol** for fair comparison. A new subgraph is solidified after one successful encounter. The consistency-based Feasibility Dominance already filters incorrect candidates before solidification.
>
> ---
> ### **Limitations**
> > Limitations is missing; what happens when induced skills encode incorrect logic that gets solidified?
>
> - **Primary failure mode: insufficient demo coverage.** Branches absent from demonstrations are missing in the offline-induced program and must be recovered by online honing. Per Q3, this gap narrows quickly (1-demo: 87.3% vs ASI 70.6%; 2-demo saturates at 93.5%).
> - **Incorrect solidified logic is contained.** Feasibility Dominance rejects failing candidates. Persisting errors remain human-readable and localizable in the FOL graph. We will add a dedicated limitations section.
>
> ---
> We believe these additional analyses and architectural breakdowns provide a more rigorous and transparent foundation for understanding NSI’s performance and design. We genuinely appreciate the opportunity to clarify and look forward to further discussion should you have any remaining questions.

---

> > ### Author Rebuttal · Reviewer_Bbky · 2026-04-04
> >
> > Thank you for the response. I have one comment though:
> > About the Limitation: Saying errors are "contained" and "human-readable" is reassuring but not fully convincing without data on how often incorrect logic actually gets solidified in practice.
> >
> > I will icnrease the score.

---

> > > ### Author Response · Authors · 2026-04-04
> > >
> > > We thank the reviewer for the continued engagement and the positive assessment of our rebuttal. Your constructive feedback has been instrumental in strengthening the manuscript, and we are grateful for your decision to increase the score.
> > >
> > >
> > > Regarding the remaining point on how often incorrect logic is actually promoted during honing: this is an important question, and we do have quantitative evidence from our rebuttal analysis. Across three runs (also reported in our Response to Reviewer rQr9, Q4), the online honing process creates tentative branches (6±2 on ALFWorld, 30±4 on TextCraft), of which **only a subset pass validation and are promoted** (3±1 and 10±2, respectively), while a smaller portion is explicitly pruned (1±1 and 6±2); the remainder stay tentative rather than being committed. Consistent with the skill-SR improvement after honing reported in our W3 response (76.1%→88.8% on ALFWorld, 57.8%→71.2% on TextCraft), **these counts indicate that honing applies a conservative and reliable promotion gate rather than automatically solidifying new logic**. We agree this deserves to be stated more clearly and will add it to the limitations section in the revision.
> > >
> > > ---
> > >
> > > We thank the reviewer for the valuable discussion. These quantitative details will be included in our final revision to strengthen the paper, and we sincerely appreciate the positive assessment.

---

### Decision · Program_Chairs · 2026-04-30

**Decision:**

Accept (regular)

**Comment:**

This paper introduces Neuro-Symbolic Skill Induction (NSI), a framework that improves long-horizon planning for LLM agents by converting interaction traces into modular, logic-grounded programs. Unlike previous methods that rely on state-blind scripts, NSI utilizes First-Order Logic to synthesize explicit control flows and dynamic variable bindings through offline induction and online evolution. Strengths include a compelling conceptual shift toward programmatic skills, impressive performance gains in survival analysis of up to 53 steps, and consistent results across multiple benchmarks such as ALFWorld and WebShop. However, the work originally faced weaknesses regarding the lack of reported variance, a heavy reliance on manually engineered domain-specific predicates, and insufficient discussion of its positioning relative to classical inductive logic programming.

The authors addressed major empirical concerns by providing three-run statistics that demonstrate low variance and consistent performance gains across all benchmarks. They also provided cross-backbone verification using Claude, Gemini, and DeepSeek to demonstrate the framework's robustness across different LLMs. Regarding the scalability of manual predicates, the authors presented quantitative evidence that a small seed set enables automated discovery of the remaining predicates with 100% recall. While the rebuttal clarified the differences between NSI and classical ILP regarding search guidance and execution regimes, some skepticism remains regarding whether the search space is truly too large for traditional symbolic methods in these specific task settings.

I am slightly leaning toward acceptance (weak accept); however, if there is no room in the program, I would not mind if my recommendation was bumped down.